# Learn from the Past: A Proxy based Adversarial Defense Framework to Boost Robustness

## Abstract

In light of the vulnerability of deep learning models to adversarial samples and the ensuing security issues, a range of methods, including Adversarial Training (AT) as a prominent representative, aimed at enhancing model robustness against various adversarial attacks, have seen rapid development. However, existing methods essentially assist the current state of target model to defend against parameter-oriented adversarial attacks with explicit or implicit computation burdens, which also suffers from unstable convergence behavior due to inconsistency of optimization trajectories. Diverging from previous work, this paper reconsiders the update rule of target model and corresponding deficiency to defend based on its current state. By introducing the historical state of the target model as a proxy, which is endowed with much prior information for defense, we formulate a two-stage update rule, resulting in a general adversarial defense framework, which we refer to as 'LAST' (**L**earn from the **P**ast). Besides, we devise a Self Distillation (SD) based defense objective to constrain the update process of the proxy model without the introduction of larger teacher models. Experimentally, we demonstrate consistent and significant performance enhancements by refining a series of single-step and multi-step AT methods (e.g., up to $\mathbf{9.2}\%$ and $\mathbf{20.5}\%$ improvement of Robust Accuracy (RA) on CIFAR10 and CIFAR100 datasets, respectively) across various datasets, backbones and attack modalities, and validate its ability to enhance training stability and ameliorate catastrophic overfitting issues meanwhile.

## 1 Introduction

Amidst the rapid development of deep learning models and their widespread deployment in real-world applications (Krizhevsky et al., 2012; Jian et al., 2016), there is a growing recognition of the vulnerability of these models to the imperceptible adversarial perturbation in input data (Kurakin et al., 2018; Carlini & Wagner, 2017). The introduction of perturbed adversarial samples can lead to the model producing specified or alternative erroneous predictions, thus jeopardizing the functionality of real-world surveillance (Dai et al., 2018), autonomous driving systems (Szegedy et al., 2013), and giving rise to critical safety concerns. Consequently, the enhancement of model robustness against adversarial samples generated by various attacks has emerged as a focal research topic in the current landscape (Papernot et al., 2016; Chen et al., 2020; Latorre et al., 2023).

While various defense methods (Zhang et al., 2019; Dong et al., 2020) have been investigated to mitigate adversarial attacks, Adversarial Training (AT) (Madry et al., 2017; Shafahi et al., 2019) is widely acknowledged as among the most efficacious strategies, of which the essence lies in addressing the min-max optimization problem. Under this Standard AT (SAT) formulation, different adversarial attacks (Rebuffi et al., 2022; Yuan et al., 2021) could be incorporated to improve the attack process of the attacked model (i.e., target model), including the single-step attack (Goodfellow et al., 2014) based and multi-step attack based AT (Madry et al., 2017). As for the defense process, various factors such as the perturbation sizes and the data quality always lead to the unstable convergence behavior of target model (Dong et al., 2022). In particular, catastrophic overfitting (Li et al., 2020) refers to significant performance decrease during the training process when trained with larger perturbation, which severely limits the robustness improvement of target model when trained under larger perturbation sizes. On top of that, several lines of works have explored heuristic defense techniques to enhance the defense process, including introducing additional robust teacher models (Pang et al., 2020) and designing specialized regularization terms (Andriushchenko & Flammarion, 2020). Whereas, (*i*) these

methods essentially introduce additional prior knowledge or design complex learning strategies with explicit or implicit computation cost (e.g., introducing regularization constraints online or pretrained teacher models offline). Besides, *(ii)* they have always spared efforts to assist the current state of target model itself to defend the parameter-oriented attack, which always suffers from inconsistency among the historical states, and leads far too easily to unstable convergence behavior.

## 1.1 CONTRIBUTIONS

In this paper, we do not follow the SAT process to use the target model to directly respond to the generated adversarial example, and reconsider the update paradigm of defense model from the perspective of its optimization trajectories. Specifically, we adopt the historical parameter state of the target model denoted as the proxy model, and design a two-stage update rule to construct a general adversarial defense framework, termed LAST (**L**earn from the **Past**). During the defense process, we first perform gradient descent to update the proxy model to estimate the next state to defend the parameter-oriented attack, and then employ the estimated state and current state of target model to calculate the different unit for update of target model. At the second stage, we update the proxy model and target model with the current state and differential unit as the update direction, respectively. Furthermore, we propose a new Self Distillation (SD) based defense objective to regularize the update process of proxy model without introducing additional teacher models, which effectively alleviates the catastrophic overfitting problem.

Experimentally, we demonstrate the effectiveness and consistent performance improvement of LAST framework by improving four single-step and multi-step AT methods based on various datasets and commonly used backbones, which also verify its ability to stabilize the training and alleviates the catastrophic overfitting problem. Especially, in Fig. 1, we plot the adversarial loss landscape (Liu et al., 2020) of four original SAT methods and the corresponding improved versions trained using PreActResNet18 with $\epsilon = 8/255$. The adversarial loss is calculated with $\mathcal{L}_{\texttt{atk}}(\mathbf{I} + x\vec{\iota} + y\vec{o})$, where $\mathbf{I}$ denotes the original image from CIFAR10 dataset, $\vec{\iota} = \texttt{sgn}(\nabla_{\mathbf{I}}\mathcal{L}_{\texttt{atk}}(\mathbf{I}))$ and $\vec{o} \sim \text{Rademacher}(0, 0.5)$ are the sign gradient direction and random direction ($x$ and $y$ are the corresponding linear coefficients). As it can be observed, the models trained with LAST framework exhibit lower loss, smoother landscapes and smaller loss gaps within the range of surfaces plotted in the subfigure, which validates the significant robustness improvement of the proposed adversarial defense framework.

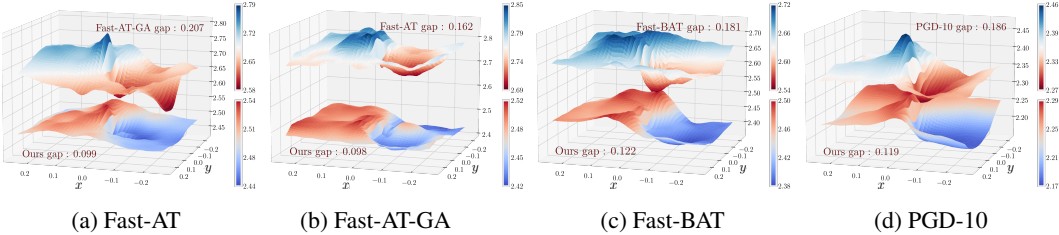

| (a) Fast-AT | (b) Fast-AT-GA | (c) Fast-BAT | (d) PGD-10 |

Figure 1: The four subfigures visualize the adversarial loss landscape w.r.t. input variations of four original SAT methods and corresponding improved version with LAST framework. We also report the gap of maximum and minimum losses within the range of $x, y \in [-0.25, 0.25]$. As it can be observed evidently, the models trained with LAST framework exhibit lower loss, smoother loss landscapes along with smaller loss gaps within the perturbation range.

We summarize our main contributions as follows.

1. As one of the most significant features, this paper stands as the first of its kind to revisit the defense update rule of SAT process and its deficiency from the perspective of its optimization trajectories, which also emphasizes the importance of the historical states of target model to help defend against the parameter-oriented adversarial attack.

2. By introducing the historical state of the target model as its proxy, we construct a simple but much effective two-stage adversarial defense framework, named LAST, endowed with great potential to serve as an alternative of SAT and consistently improve existing methods to boost robustness with almost no additional cost.

3. Based on the proxy model, we design a SD defense objective to constrain the learning process of proxy model without requirements of pretrained teacher models. The new defense objective (along with the new update rule) could be flexibly integrated into SAT methods to stabilize the training process and alleviate the catastrophic overfitting problem.

4. We implement the LAST framework based on various SAT methods, and verify its consistent performance improvement (e.g., up to **9.2**% and **20.5**% increase of RA compared with PGD-AT under AutoAttack ($\epsilon = 16.255$) on CIFAR10 and CIFAR100 datasets, respectively) with different backbones, datasets, attack modalities, and also demonstrate its ability to enhance training stability and ameliorate overfitting issues.

More detailed related works on the adversarial attack and defense could be found in Appendix. A.1. In Sec. 2, we first review the SAT process, and then proposed our LAST framework along with the Self Distillation (SD) defense objective. Note that we also provide comprehensive analysis on the effectiveness of LAST framework and differences from previous techniques in Sec. 2.4. In Sec. 3, we conduct extensive experiments and analyze the training convergence behavior by consistently improving various SAT methods. The detailed hyperparamater settings and descriptions of baselines could be found in Appendix A.2. Last but not least, we provide ablation studies and more comparative results in Appendix A.3 and A.4.

## 2 A Proxy based Two-Stage Adversarial Defense Framework

In this section, we first review the general formulation about the SAT process. Based on its deficiency during the defense process, we propose to introduce the historical state of target model as its proxy, and construct a proxy-based two-stage adversarial defense framework. Furthermore, a new self distillation based defense objective without introducing any additional pretrained teacher models is proposed to stabilize the training process and alleviate the catastrophic overfitting problem. More discussion on the proposed update rule of LAST framework is also provided.

### 2.1 Preliminaries

To enhance the robustness of deep learning model, SAT has been thoroughly evaluated and regarded as one of the most effective adversarial defense strategies. Generally speaking, SAT could be formulated as the min-max optimization problem (Madry et al., 2017), where the attack model aims to maximize the objective by injecting imperceptible adversarial perturbation to the original input, while the defense model (i.e., target model) optimizes the parameters with gradient descent to stay robust against the perturbation. The attack and defense objectives for this problem are usually defined as the same form. Here we first define the training dataset and input data pair as $\mathcal{D} = \{\boldsymbol{u}_i, \boldsymbol{v}_i\}_{i=1}^{\mathcal{M}}$, and denote the target model as $\mathcal{T}_{\boldsymbol{\theta}}$, where $\boldsymbol{\theta}$ are parameters of the target model. Then a general-purpose SAT formulation could be written as follows

$$\min_{\boldsymbol{\theta}} \mathbb{E}_{\{\boldsymbol{u}_i, \boldsymbol{v}_i\} \in \mathcal{D}} \left[ \max_{\boldsymbol{\delta} \in \mathcal{S}} \mathcal{L}_{\texttt{atk}}\big(\mathcal{T}_{\boldsymbol{\theta}}(\boldsymbol{u}_i + \boldsymbol{\delta}), \boldsymbol{v}_i\big) \right], \tag{1}$$

where $\boldsymbol{\delta}$ is the perturbation subject to the constraint $\mathcal{S} = \{\boldsymbol{\delta} \mid \|\boldsymbol{\delta}\|_{\rho} \leq \epsilon\}$ with $\epsilon$-toleration $\rho$ norm, and $\mathcal{L}_{\texttt{atk}}$ denotes the attack objective. Typically, $\boldsymbol{\delta}$ is generated by $K$-step maximization of the attack objective following

$$\boldsymbol{\delta}_{k+1} \leftarrow \Pi_{\epsilon}\big(\boldsymbol{\delta}_k + \boldsymbol{\alpha} \cdot \texttt{sgn}\nabla_{\boldsymbol{\delta}}\mathcal{L}_{\texttt{atk}}(\mathcal{T}_{\boldsymbol{\theta}}(\boldsymbol{u}_i + \boldsymbol{\delta}), \boldsymbol{v}_i)\big), \ k = 0, 1, \cdots, K-1. \tag{2}$$

where $\Pi$ and sgn are the projection and element-wise $sign$ operation. $\boldsymbol{\delta}_0$ is uniformly initialized from $(-\epsilon, \epsilon)$. When $K$ is set as $K = 1$ or $K > 1$, we could derive two major types of adversarial attacks, i.e., FGSM and PGD attacks. These perturbation generated online according to the current state of target model $\boldsymbol{\theta}_i$ is actually parameter-oriented to a great extent. As for the SAT process, the target model improves its robustness by performing gradient descent according to the attack objective by

$$\boldsymbol{\theta}_{i+1} = \boldsymbol{\theta}_i - \nabla_{\boldsymbol{\theta}}\mathcal{L}_{\texttt{atk}}\big(\mathcal{T}_{\boldsymbol{\theta}}\left(\boldsymbol{u}_i + \boldsymbol{\delta}_K\right), \boldsymbol{v}_i\big). \tag{3}$$

In this paper, we do not focus on the adversarial attack process of this min-max optimization problem, but turn our foresight to how the target model reacts to these adversarial examples. It has been discussed (Nakkiran et al., 2021) before that the target model trained with SAT always

suffers from unstable convergence behavior and even more severe problems such as catastrophic overfitting phenomenon due to various factors, e.g., the size of target models, the perturbation radius and data quality causing label noise in the perturbed data pairs. Moreover, the sgn operation and constraint of input images (i.e., $[0, 1]$) also introduce much bias to the gradient-based defense of target model. From this perspective, when faced with adversarial examples w.r.t. the current state of target model, it is always too hard for the target model to capture the attack modality and the correspondence between $\delta_K$ and $\theta_i$. On top of that, updating $\theta_i$ along the gradient descent direction of $\mathcal{L}_{\text{atk}}\left(\mathcal{T}_\theta\left(u_i + \delta_K\right), v_i\right)$ based on this generated parameter-oriented perturbation in Eq. (3), unintentionally leads to significant inconsistency among the optimization trajectories, i.e., $\{\cdots, \theta_i - \theta_{i-1}, \theta_{i+1} - \theta_i, \cdots\}$, and exacerbates the unstable training process meanwhile.

From this new perspective to understand the update rule of SAT, several lines of works have been made, explicitly or implicitly, to modify the process of updating target models, i.e., adding regularization terms to optimize new forms of defense objectives (Andriushchenko & Flammarion, 2020), introducing pretrained teacher model to correct the labels for supervision (Dong et al., 2022), estimations of hyper gradient through Bilevel Optimization(BLO) reformulation (Zhang et al., 2022). To summarize, (*i*) these methods have spared efforts to reconsider the influence of training data, forms of defense objective and coupled relationship of SAT formulation to introduce extra prior or design complex learning strategies along with additional computation cost (increased runtime online or offline). Besides, (*ii*) they follow the commonly used criterion to assist the current state of target model $\theta_i$ itself to defend the adversarial perturbation $\delta_K$, which is ineffective to maintain the consistency among the optimization trajectories, and easily causes unstable convergence behavior. Therefore, we pose the following inquiry: *is there a more effective response of the target model to the parameter-oriented attacks?* In the next subsection, we reconsider the update rule of defense model from the perspective of its optimization trajectories, and proposed to reuse the historical state of the target model to construct a new adversarial defense framework.

## 2.2 ENHANCE ROBUSTNESS WITH THE LAST FRAMEWORK

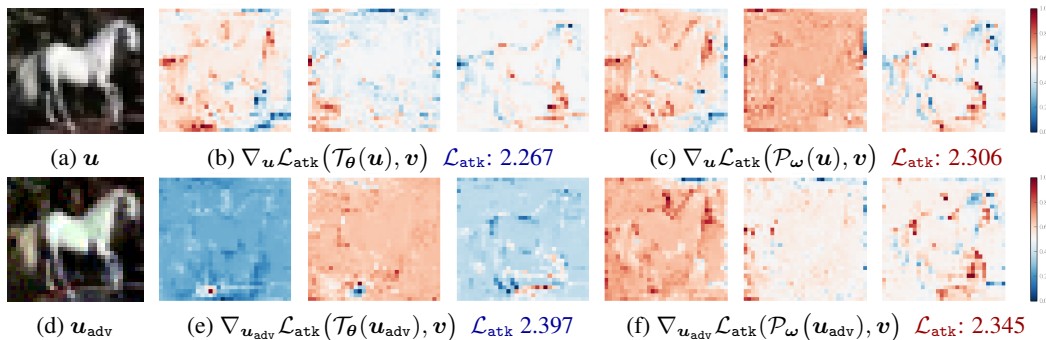

|   |   |   |
|---|---|---|
| (a) $u$ | (b) $\nabla_u \mathcal{L}_{\text{atk}}\left(\mathcal{T}_\theta(u), v\right)$ $\mathcal{L}_{\text{atk}}$: 2.267 | (c) $\nabla_u \mathcal{L}_{\text{atk}}\left(\mathcal{P}_\omega(u), v\right)$ $\mathcal{L}_{\text{atk}}$: 2.306 |
| (d) $u_{\text{adv}}$ | (e) $\nabla_{u_{\text{adv}}} \mathcal{L}_{\text{atk}}\left(\mathcal{T}_\theta(u_{\text{adv}}), v\right)$ $\mathcal{L}_{\text{atk}}$ 2.397 | (f) $\nabla_{u_{\text{adv}}} \mathcal{L}_{\text{atk}}\left(\mathcal{P}_\omega\left(u_{\text{adv}}\right), v\right)$ $\mathcal{L}_{\text{atk}}$: 2.345 |

Figure 2: We illustrate the heat map of input gradient w.r.t. $u$ and $u_{\text{adv}} = u + \delta_K$, where $\delta_K$ is generated by attacking $\mathcal{T}_\theta$ with PGD-10, $\epsilon = 8/255$. The three columns of subfigure (b), (c), (e) and (f) correspond to the input gradient for three channels normalized to $[0, 1]$. It can be observed that the input gradients of $\mathcal{P}_\omega$ exhibit less growth of loss (i.e., (c) $\rightarrow$ (f)) and generates more salient input gradient w.r.t. $u_{\text{adv}}$ around the shape of the horse compare with $\mathcal{T}_\theta$ (i.e., (b) $\rightarrow$ (e)).

As it is summarized before, the perturbation is generated according to the current state of the target model parameters throughout the attack process, which makes the attack parameter-oriented in essence. To verify this hypothesis, we first generate the adversarial example, i.e., $u_{\text{adv}} = u + \delta_K$, where $\delta_K$ targets at the best model trained with early stopping, denoted as $\mathcal{T}_\theta$. Then we use proxy model to represent the historical state of target model $\mathcal{T}_\theta$, denoted as $\mathcal{P}_\omega$, and use $x_{\text{adv}}$ to attack both $\mathcal{T}_\theta$ and $\mathcal{P}_\omega$. Generally speaking, $\mathcal{T}_\theta$ obtained with early stopping has definitely stronger robustness than $\mathcal{P}_\omega$. In Fig. 2, we illustrate the heatmap of input gradient to analyze how the target model and its proxy model react to $u_{\text{adv}}$. The input gradient of an image represents how sensitive the model is to changes in the pixel values of this image (Chan et al., 2020), and the output of robustly trained model will generate salient input gradients which resemble the clean image. When faced with the parameter-oriented attack, it is shown in subfigure (e) that the output of $\mathcal{T}_\theta$ is seriously degraded and

no longer produce salient input gradient in each channel. In comparison, $\mathcal{P}_{\boldsymbol{\omega}}$ is more robust against this parameter-oriented adversarial example and has salient gradient around these pixels which matter most to the model decision in subfigure (f).

Therefore, although the target model shows great vulnerability to this perturbation, *the historical states of the target model and its gradient information is inaccessible to the attack model, and is of great value to provide prior information for the adversarial defense*. Furthermore, this hypothesis could also be essentially verified by the phenomenon that the target model consistently exhibits superior performance when subjected to transfer-based black-box attacks compared to white-box attacks of equivalent intensity.

Inspired by this principle, we make the first attempt to introduce the historical states of target model to estimate better response to the parameter-oriented attack w.r.t. the current state of target model. In detail, we first define the last state of target model as its proxy, i.e., $\boldsymbol{\omega}_i = \boldsymbol{\theta}_{i-1}, i = 1, \cdots, \mathcal{M}$, where $\boldsymbol{\omega}_0$ is initialized using $\boldsymbol{\theta}_0$. In the following, we use $\mathcal{L}_{\texttt{def}}$ to represent the defense objective. During the attack process, we adopt the same scheme as SAT to generate the adversarial perturbation, i.e., $\boldsymbol{\delta}_K$. As for the defense strategy, we first perform gradient descent with $\mathcal{P}_{\boldsymbol{\omega}}$ according to $\mathcal{L}_{\texttt{def}}(\mathcal{P}_{\boldsymbol{\omega}}(\boldsymbol{u}_i + \boldsymbol{\delta}_K), \boldsymbol{v}_i)$ to estimate the next state of target model, which could be described as

$$\tilde{\boldsymbol{\omega}} = \boldsymbol{\omega}_i - \boldsymbol{\beta} \cdot \nabla_{\boldsymbol{\omega}} \mathcal{L}_{\texttt{def}}\big(\mathcal{P}_{\boldsymbol{\omega}_i}(\boldsymbol{u}_i + \boldsymbol{\delta}_K),\ \boldsymbol{v}_i\big), \tag{4}$$

where $\boldsymbol{\beta}$ denotes the learning rate of $\mathcal{P}_{\boldsymbol{\omega}}$. Then we employ $\tilde{\boldsymbol{\omega}}$ and current state of target model (i.e., $\boldsymbol{\theta}_i$) to calculate the differential unit $\mathcal{G}_{\boldsymbol{\theta}}$ as the update direction, which is denoted as $\mathcal{G}_{\boldsymbol{\theta}} = \boldsymbol{\theta}_i - \tilde{\boldsymbol{\omega}}$. For the second stage, we update $\boldsymbol{\omega}_i$ to record the current state of target model, and then perform gradient descent of $\boldsymbol{\theta}_i$ with $\mathcal{G}_{\boldsymbol{\theta}}$. The whole adversarial defense framework including the attack and two-stage defense update rule is described in Alg. 1. The step size of $\boldsymbol{\theta}_i$, i.e., $\boldsymbol{\gamma}$, will be discussed further in Sec 2.4. This new update rule is supposed to generated better response which is more robust to defend against this parameter-oriented attack. In the next subsection, we introduce constraints to the update of proxy model to estimate $\tilde{\boldsymbol{\omega}}$ inspired by the self distillation idea which helps stabilize the training and alleviate the catastrophic overfitting problem.

---

**Algorithm 1** The Proposed LAST Framework

---

**Input:** Training epochs $\mathcal{J}$, $\mathcal{M}$ batches of data pairs $(\boldsymbol{u}_i, \boldsymbol{v}_i)$, attack iteration $K$, target model $\mathcal{T}_{\boldsymbol{\theta}}$ parameterized by $\boldsymbol{\theta}$, and proxy model $\mathcal{P}_{\boldsymbol{\omega}}$ parameterized by $\boldsymbol{\omega}$, perturbation range $\boldsymbol{\epsilon}$.
1: *// Initialize the proxy model $\mathcal{P}_{\boldsymbol{\omega}}$.*
2: $\boldsymbol{\omega}_0 = \boldsymbol{\theta}_0$.
3: **for** $j = 0 \rightarrow \mathcal{J} - 1$ **do**
4:     **for** $i = 0 \rightarrow \mathcal{M} - 1$ **do**
5:         Initialize $\delta_0$.
6:         *// Generate the perturbation with target model $\mathcal{T}_{\boldsymbol{\theta}}$.*
7:         **for** $k = 0 \rightarrow K - 1$ **do**
8:             $\boldsymbol{\delta}_{k+1} = \boldsymbol{\delta}_k + \boldsymbol{\alpha} \cdot \texttt{sgn}\big(\nabla_{\boldsymbol{\delta}} \mathcal{L}_{\texttt{atk}}(\mathcal{T}_{\boldsymbol{\theta}_i}(\boldsymbol{u}_i + \boldsymbol{\delta}_k),\ \boldsymbol{v}_i)\big)$. ($\boldsymbol{\alpha}$ denotes the attack step size)
9:             $\boldsymbol{\delta}_{k+1} = \max\big[\min(\boldsymbol{\delta}_{k+1}, \boldsymbol{\epsilon}), -\boldsymbol{\epsilon}\big]$.
10:         **end for**
11:         *// Stage 1: Estimate update direction of $\boldsymbol{\theta}_i$ to defend.*
12:         $\tilde{\boldsymbol{\omega}} = \boldsymbol{\omega}_i - \boldsymbol{\beta} \cdot \nabla_{\boldsymbol{\omega}} \mathcal{L}_{\texttt{def}}\big(\mathcal{P}_{\boldsymbol{\omega}_i}(\boldsymbol{u}_i + \boldsymbol{\delta}_K),\ \boldsymbol{v}_i\big)$.   ($\boldsymbol{\beta}$ denotes the learning rate of $\mathcal{P}_{\boldsymbol{\omega}}$)
13:         $\mathcal{G}_{\boldsymbol{\theta}} = \boldsymbol{\theta}_i - \tilde{\boldsymbol{\omega}}$.   (Compute the differential unit $\mathcal{G}_{\boldsymbol{\theta}}$)
14:         *// Stage 2: Update $\boldsymbol{\omega}_i$ and $\boldsymbol{\theta}_i$ sequentially.*
15:         $\boldsymbol{\omega}_{i+1} = \boldsymbol{\theta}_i$.
16:         $\boldsymbol{\theta}_{i+1} = \boldsymbol{\theta}_i - \boldsymbol{\gamma} \cdot \mathcal{G}_{\boldsymbol{\theta}}$.   ($\boldsymbol{\gamma}$ denotes the learning rate of $\mathcal{T}_{\boldsymbol{\theta}}$)
17:     **end for**
18: **end for**

---

### 2.3 SELF DISTILLATION BASED DEFENSE OBJECTIVE

Based on the introduced proxy model, which captures the historical states to introduce prior information for defense (Step 12 in Alg. 1), we further delve into the defense objective to constrain the learning process of proxy model and alleviate the overfitting problem. As it is shown in Fig. 2, although $\mathcal{T}_{\boldsymbol{\theta}}$ is less sensitive to the adversarial attack targeted at $\boldsymbol{\theta}$, the perturbation still deteriorates

the output of $\mathcal{T}_{\boldsymbol{\theta}}(\boldsymbol{u}_{\mathrm{adv}})$ which may lead to misclassification. Whereas, the direct output of target model, which refers to the soft targets in Knowledge Distillation (KD) (Li, 2018), also reflects which part the target model concerns about. When faced with the clean image and adversarial perturbation, the proxy model is supposed to generate outputs that have more similar distributions. Unlike these methods generating supervised soft targets with a larger teacher model, we propose to constrain the estimation of $\tilde{\boldsymbol{\omega}}$ with the distance between soft targets of clean image and the corresponding adversarial image. Here we denote the temperature as $\boldsymbol{\tau}$, then the proposed defense objective could be written as follows

$$\mathcal{L}_{\mathrm{def}} = (1 - \boldsymbol{\mu}) \cdot \mathcal{L}_{\mathrm{atk}}\big(\mathcal{P}_{\boldsymbol{\omega}}(\boldsymbol{u}_{\mathrm{adv}}), \boldsymbol{v}\big) + \boldsymbol{\mu} \cdot \mathcal{L}_{\mathrm{KL}}\big(\mathcal{P}_{\boldsymbol{\omega}}(\boldsymbol{u}_{\mathrm{adv}})/\boldsymbol{\tau}, \mathcal{P}_{\boldsymbol{\omega}}(\boldsymbol{u})/\boldsymbol{\tau}\big), \tag{5}$$

where $\boldsymbol{\mu} \in [0, 1)$ is the distillation coefficient to balance two loss terms, and $\mathcal{L}_{\mathrm{KL}}$ denotes the Kullback-Leibler (KL) divergence (Kullback & Leibler, 1951) to measure the distance between two distributions of the soft targets. In this way, the proxy model is supposed to behave as consistently as possible when faced with clean or adversarial examples and generate correct classification results meanwhile. Moreover, the introduced defense objective supervises the learning process of proxy model without introducing (larger) pretrained teacher models or additional updates of models, thus can be flexibly integrated to the proposed algorithmic framework in Alg. 1 at the least computational cost. In the experimental part, we demonstrate the effectiveness of LAST framework along with the SD defense objective which stabilizes the training and alleviates the catastrophic overfitting problem.

## 2.4 DISCUSSION ON THE PROXY BASED UPDATE RULE

Here we provide more discussion and different perspectives to analyze the effectiveness of the introduced proxy model and two-stage update rule. With a simple substitution and deformation to Step. 16, we could derive $\boldsymbol{\theta}_{i+1} - \boldsymbol{\theta}_i = -\boldsymbol{\gamma} \cdot \mathcal{G}_{\boldsymbol{\theta}} = \boldsymbol{\gamma} \cdot (\tilde{\boldsymbol{\omega}} - \boldsymbol{\theta}_i)$, where $\boldsymbol{\gamma}$ denotes the learning rate of $\mathcal{T}_{\boldsymbol{\theta}}$. It can be observed that the historical sequences of $\boldsymbol{\theta}_i$ is always constrained by the estimation of distance between $\tilde{\boldsymbol{\omega}}$ and $\boldsymbol{\theta}_i$, both of which is derived from $\boldsymbol{\theta}_{i-1}$. Assume that the target model of the critical state unexpectedly diverges, Eq. (4) could estimate $\tilde{\boldsymbol{\omega}}$ which is more robust against this parameter-oriented perturbation to assist the updates of $\boldsymbol{\theta}_i$. This update rule is supposed to improve the consistency between adjacent states of the target model as the target model converges, i.e., $\{\cdots, \boldsymbol{\theta}_i - \boldsymbol{\theta}_{i-1}, \boldsymbol{\theta}_{i+1} - \boldsymbol{\theta}_i, \cdots\}$.

Besides, we could describe the update format as $\boldsymbol{\theta}_{i+1} = (1 - \boldsymbol{\gamma}) \cdot \boldsymbol{\theta}_i + \boldsymbol{\gamma} \cdot \tilde{\boldsymbol{\omega}}$, where $\tilde{\boldsymbol{\omega}}$ serve as estimated response generated by $\boldsymbol{\theta}_{i-1}$ to defend the adversarial example targeted at $\boldsymbol{\theta}_i$. On top of that, $\boldsymbol{\gamma}$ is the aggregation coefficient to balance the influence of responses to historical attacks and current attacks. The format of this update rule is similar to these momentum based optimizers (Sutskever et al., 2013) to some extent, which refer to the accumulation of historical gradients to perform gradient descent. Furthermore, we could also find evidence to demonstrate the effectiveness of this update rule from other techniques such as Stochastic Weight Averaging (SWA) (Izmailov et al., 2018), which smoothes the weights by averaging multiple checkpoints along the training process. This technique have been demonstrated to be effective to find flatter solutions than SGD, and applied in various applications (Athiwaratkun et al., 2018; Yang et al., 2019). Specifically, the weights of SWA are simply accumulated by the exponential weighted average of the historical weights. In comparison, the new update rule *combines the response of proxy model to the parameter-oriented attack, which bridge the historical states and current states to improve consistency among the optimization trajectories and introduce extra prior for defense*. Therefore, the introduced proxy model is of great significance and cannot be simply replaced by using momentum-like optimizers or the stochastic averaging of weights by the SWA technique.

## 3 EXPERIMENTS

In this section, we first demonstrate the robustness improvement of LAST framework based on popular single-step and multi-step methods in two subsections, respectively. We also compare the loss landscape and convergence behavior of test robust loss, and RA to verify its stronger stability and defense capability against larger adversarial perturbation. Finally, we also analyze the defense performance of the proposed framework under transfer-based black box attacks. Note that we provide the basic experimental settings for different AT methods, datasets and models used for robustness evaluation in Appendix A.2 due to limited space. And more ablation results of hyperparameters and full results are provided in Sec. A.3 and Sec. A.4.

## 3.1 EVALUATION WITH SINGLE-STEP AT METHODS

Table 1: We report the SA and RA of Fast-AT, Fast-AT-GA and Fast-BAT under PGD attack (PGD-10 and PGD-50) and AutoAttack. We use $m\pm n$ to denote the mean SA (i.e., $m$) with standard deviation (i.e., $n$) by running all the algorithms with 3 random seeds.

| CIFAR-10 dataset, PARN-18 trained with $\epsilon = 8/255$ | | | | |
|---|---|---|---|---|
| Method | PGD-10 (%) | | PGD-50 (%) | |
| | $\epsilon = 8/255$ | $\epsilon = 16/255$ | $\epsilon = 8/255$ | $\epsilon = 16/255$ |
| Fast-AT | 47.03±0.29 | 13.79±0.15 | 44.94±0.52 | 8.85±0.20 |
| LF-AT(Ours) | **47.17**±0.15 | **14.48**±0.23 | **45.50**±0.04 | **9.89**±0.14 |
| Fast-AT-GA | 48.30±0.13 | 16.36±0.14 | 46.63±0.33 | 11.12±0.12 |
| LF-AT-GA (Ours) | **48.60**±0.06 | **17.52**±0.02 | **47.25**±0.09 | **12.63**±0.17 |
| Fast-BAT | 50.42±0.36 | 18.29±0.18 | 49.07±0.39 | 13.31±0.16 |
| LF-BAT (Ours) | **50.65**±0.19 | **19.73**±0.05 | **49.66**±0.20 | **15.25**±0.20 |
| Method | SA (%) | AutoAttack (%) | | Time |
| | | $\epsilon = 8/255$ | $\epsilon = 16/255$ | (Sec/ Iteration) |
| Fast-AT | **83.56**±0.06 | 41.80±0.68 | 7.32±0.27 | $5.543 \times 10^{-2}$ |
| LF-AT (Ours) | 81.70±0.15 | **42.11**±0.19 | **8.13**±0.20 | $5.719 \times 10^{-2}$ |
| Fast-AT-GA | **81.00**±0.59 | 43.17±0.21 | 9.04±0.18 | $1.632 \times 10^{-1}$ |
| LF-AT-GA (Ours) | 79.18±0.13 | **43.31**±0.23 | **10.22**±0.05 | $1.643 \times 10^{-1}$ |
| Fast-BAT | **82.01**±0.04 | 45.51±0.44 | 10.98±0.19 | $1.644 \times 10^{-1}$ |
| LF-BAT (Ours) | 79.72±0.14 | **45.54**±0.27 | **12.23**±0.27 | $1.656 \times 10^{-1}$ |

In this subsection, we first evaluate the Standard Accuracy (SA) and RA of Fast-AT, Fast-AT-GA, Fast-BAT and our improved versions trained on the CIFAR10 dataset using PARN-18 backbone with $\epsilon = 8/255$ in Tab. 1. It can be observed that the LAST framework shows consistent performance improvement of RA on PGD-10, PGD-50 and AutoAttack. In particular, the target models trained with LAST framework are significantly more robust when faced with unknown adversarial attacks of larger perturbation size (test with $\epsilon = 16/255$). Furthermore, we evaluate the average runtime of SAT methods and our improve ones for each iteration. It can be observed that improving existing AT methods with our framework only slightly increases the runtime, which demonstrates its potential to serve as an alternative of SAT with almost no additional computation cost. Besides, the adversarial landscapes in the subfigure (a)-(c) of Fig. 1 also show that combining our update rule will generate smoother adversarial loss surfaces with the smaller loss gap, which make the model stay more robust when faced with adversarial input under different perturbation sizes and noise levels.

Table 2: Illustrating the SA and RA of Fast-AT, Fast-AT-GA and Fast-BAT under PGD attack (PGD-10 and PGD-50) and AutoAttack on CIFAR100 dataset. We use ↑ to report the average improvement of RA by running all the algorithms with 3 random seeds. More detailed results with standard deviations can be found in Tab. 5.

| CIFAR-100 dataset, PARN-18 trained with $\epsilon = 8/255$ | | | | | | |
|---|---|---|---|---|---|---|
| Method | SA (%) | PGD-10 (%) | | PGD-50 (%) | | AutoAttack (%) |
| | | $\epsilon = 8/255$ | $\epsilon = 16/255$ | $\epsilon = 8/255$ | $\epsilon = 16/255$ | $\epsilon = 16/255$ |
| Fast-AT | **55.087** | 24.330 | 7.430 | 23.533 | 5.520 | 4.153 |
| LF-AT (Ours) | 50.817 | **25.190**$_{\uparrow 0.86}$ | **9.003**$_{\uparrow 1.57}$ | **24.373**$_{\uparrow 0.84}$ | **7.497**$_{\uparrow 1.98}$ | **5.443**$_{\uparrow 1.29}$ |
| Fast-AT-GA | **53.253** | 25.660 | 8.603 | 24.853 | 6.836 | 5.320 |
| LF-AT-GA (Ours) | 48.220 | **25.887**$_{\uparrow 0.23}$ | **10.277**$_{\uparrow 1.67}$ | **25.433**$_{\uparrow 0.58}$ | **8.813**$_{\uparrow 1.97}$ | **6.270**$_{\uparrow 0.95}$ |
| Fast-BAT | **42.793** | 22.603 | 8.920 | 22.059 | 7.813 | 5.807 |
| LF-BAT (Ours) | 42.460 | **23.153**$_{\uparrow 0.55}$ | **9.840**$_{\uparrow 0.92}$ | **22.783**$_{\uparrow 0.72}$ | **8.740**$_{\uparrow 0.93}$ | **6.103**$_{\uparrow 0.30}$ |

In Tab. 2, we also compare the performance of these methods with our LAST framework trained on the CIFAR100 dataset, and report the average improvement of these methods by combining the LAST framework. Note that we calculate the mean RA and its standard deviation by running different methods with different random seeds on both CIFAR10 and CIFAR100 datasets. As it is shown, our framework exhibits the capacity to consistently enhance existing methods on the larger dataset. Besides, we also implement these methods and our LAST framework based on the larger WRN-34-10 backbone, and the detailed results could be found in Tab. 4.

Furthermore, in Fig. 3, we focus on the catastrophic overfitting phenomenon when faced with stronger adversaries by setting $\epsilon = 16/255$. It can be observed that the robustness of Fast-AT drops significantly during the training process, and its loss landscape (obtained with the best model by early stopping) shows violent fluctuations influence by the injected perturbation and random noise. When we implement Fast-AT under the LAST framework together with SD objective, the update direction are continuously corrected by the proxy model and prior information of soft targets, which finally leads to more stable convergence behavior, performance boost and also smoother loss landscape. More ablation results without the proposed SD objective can be found in the Appendix.

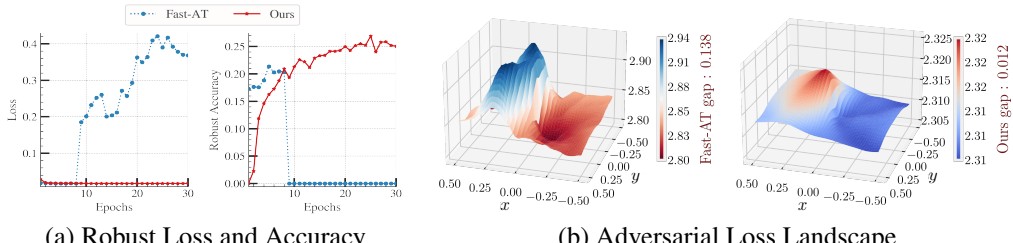

(a) Robust Loss and Accuracy        (b) Adversarial Loss Landscape

Figure 3: Subfigure (a) illustrates the convergence behavior of test loss and RA for Fast-AT and ours on CIFAR10 dataset under PGD-10 attack with $\epsilon = 16/255$. In Subfigure (b), we compare the adversarial loss landscape for Fast-AT and our improved version for comparison. Note that we follow the method described in Fig. 1 with larger scale of linear coefficients $x, y \in [-0.5, 0.5]$.

## 3.2 EVALUATION WITH MULTI-STEP AT METHODS

Table 3: We report the SA and RA of PGD-AT(-10) and our improved version under PGD attack (PGD-10 and PGD-50) and AutoAttack on CIFAR10 and CIFAR100 dataset by running both methods with 3 random seeds. More detailed results with standard deviation could be found in Tab. 6.

| Method | SA (%) | PGD-10 (%) | | PGD-50 (%) | | AutoAttack (%) |
|---|---|---|---|---|---|---|
| | | $\epsilon = 8/255$ | $\epsilon = 16/255$ | $\epsilon = 8/255$ | $\epsilon = 16/255$ | $\epsilon = 16/255$ |
| CIFAR-10 dataset, PARN-18 trained with $\epsilon = 8/255$ | | | | | | |
| PGD-AT | 81.948 | 51.923 | 20.310 | 50.757 | 15.677 | 13.093 |
| LPGD-AT (Ours) | **82.17** | **53.230**$_{\uparrow 1.31}$ | **22.203**$_{\uparrow 1.89}$ | **52.137**$_{\uparrow 1.38}$ | **17.587**$_{\uparrow 1.91}$ | **14.297**$_{\uparrow 1.20}$ |
| CIFAR-100 dataset, PARN-18 trained with $\epsilon = 8/255$ | | | | | | |
| PGD-AT | **49.457** | 25.837 | 9.980 | 25.377 | 8.749 | 6.667 |
| LPGD-AT (Ours) | 48.150 | **31.267**$_{\uparrow 5.43}$ | **14.903**$_{\uparrow 4.92}$ | **30.857**$_{\uparrow 5.48}$ | **13.573**$_{\uparrow 4.83}$ | **8.033**$_{\uparrow 1.37}$ |

To demonstrate that the LAST framework consistently and universally enhances established approaches, we extend the framework to the stronger PGD base AT methods. In Tab. 3, we present the test results of PGD-10 based AT (denoted as PGD-AT) and the improved method by LAST framework (denoted as LPGD-AT). It can be obviously seen that LPGD-AT shows significant better performance compared to the original SAT trained with PGD-10, and even slightly improves the SA on CIFAR 10 dataset. When we train both methods on CIFAR100 dataset, LPGD-AT achieves a substantial leap in performance compared with PGD-10 based AT. We attribute the reason why the LAST framework achieves more significant improvement on PGD (9.2% and 20.5% improvement in the last column

of Tab. 3) to the fact that the correspondence between the target model and this parameter-specific adversarial attack obtained by using multi-step attack steps is more difficult to characterize, thus the consistency between the update sequences of SAT will be worse, which makes the performance improvement of our method even more significant.

Besides, It can be observed in subfigure (d) of Fig. 1 that the loss landscape of our model trained with LAST framework has been rendered smoother, accompanied by a reduced disparity between its highest and lowest values. In addition, we compare the convergence behavior of robust loss and RA for PGD-AT and our LPGD-AT on both CIFAR10 and CIFAR100 datasets in Fig. 4. As it is illustrated, by improving the consistency among the historical states of model parameters, LPGD-AT exhibits more stable convergence behavior of both robust loss and accuracy, and finally gains higher performance after performing the multi-step learning rate decay twice.

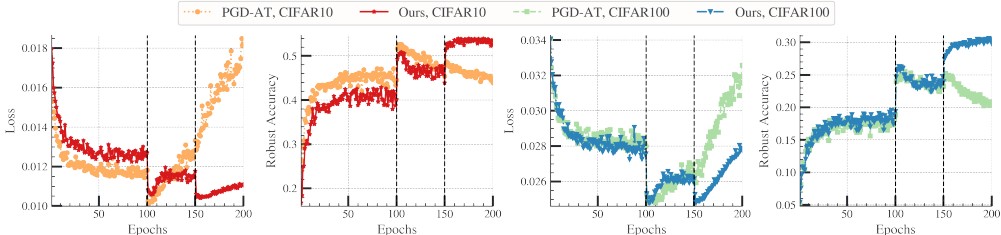

Figure 4: The first two subfigures compare the convergence behavior of test robust loss and RA trained with PGD-AT and LAST both trained with $\epsilon = 8/255$ on CIFAR10 dataset, while the left two subfigures illustrate the convergence behavior of same metrics trained on CIFAR100 dataset. The black dashed line denotes the epoch where multi-step learning rate decays.

### 3.3 EVALUATION OF GENERALIZATION PERFORMANCE

Last but not least, we also conduct analysis about the robustness of defense against black-box attacks for thorough evaluation. Practically, we plot the heatmaps of RA for different SAT methods against the transfer-based black-box adversarial attacks on CIFAR10 dataset under PGD-10 attack with $\epsilon = 8/255$ in Fig. 5. Note that the source model corresponds to the surrogate model used to generate the adversarial perturbation to attack the target models. We use F-AT and LF-AT to denote Fast-At and the improved version with LAST framework, and other methods follow the similar abbreviations. It is shown that adversarial

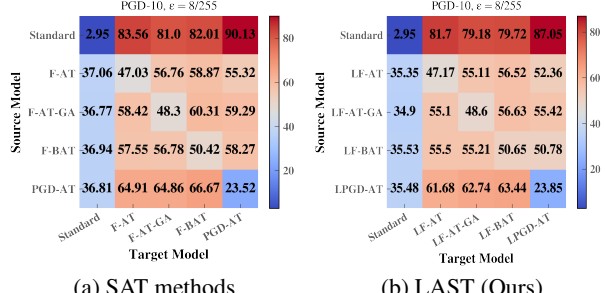

Figure 5: We visualize the heatmap of four SAT methods including Fast-AT, Fast-AT-GA, Fast-BAT, PGD-AT (i.e., 2-step PGD-AT) and their improved version under transfer-based PGD-10 attack on CIFAR10 dataset.

attacks generated based on the source models trained by LAST are more difficult to defend for standard model, and both original AT methods and our improved ones perform better under transfer-based attacks than white-box attacks.

## 4 CONCLUSION

In this study, we addressed the vulnerability of deep learning models to adversarial attacks particularly focusing on the SAT methods. Firstly, we revisit the model update process based on its optimization trajectory and introduce the historical state as proxy model, leading to the development of the novel LAST framework. We also propose the SD defense objective that doesn't rely on large pretrained teacher models. Through extensive experiments, we demonstrated LAST's consistent performance improvements across datasets, backbones, and attack scenarios, along with its ability to enhance training stability.

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

## A  APPENDIX

We first present an overview of the pertinent literature concerning adversarial attacks and adversarial defense strategies in Appendix A.1. Then we provide detailed experimental settings including the datasets, backbones, evaluation metrics, baselines and training details in Appendix A.2. In Appendix A.3, we conduct the ablation study to investigate the influence of the proposed SD defense objective and configurations of specific hyperparameters for the LAST framework. Then we implement the above methods based on the larger WRN-34-10 structure, and provide more detailed results about the evaluation of original AT methods and our proposed framework with mean RA and standard deviation in Appendix A.4. Note that we provide part of the training code in the supplementary materials, and the full repository will be released later upon acceptance.

### A.1  RELATED WORKS

**Adversarial attack.**  Generally speaking, two branches of adversarial attacks have been well explored including white-box and black-box attacks (Rebuffi et al., 2022). Here we focus on the white-box gradient-based adversarial attacks (Yuan et al., 2021), which possess full knowledge of the internal structure and parameters of the target deep learning model and leverage gradient information to craft adversarial samples. Specifically, single-step attack methods, e.g. Fast Gradient Sign Method (FGSM) (Goodfellow et al., 2014), generates adversarial examples through a single, small perturbation of input data to produce erroneous classification results or misleading outputs in a single step. The single-step attack method could be naturally extended to the multi-step version by iterative optimization with small step size, e.g., BIM (Kurakin et al., 2018) attack. Then PGD (Madry et al., 2017) attack improves BIM attack with more attack steps and random initialization of the perturbation. On top of that, AutoAttack (Croce & Hein, 2020), a specialized framework encompasses a variety of advanced adversarial attack methods and techniques such as CW (Carlini & Wagner, 2017) and DLR losses, have been wide recognized and used to evaluate the robustness of models.

**Adversarial defense.**  Different branches of adversarial defense methods (Zhang et al., 2019; Dong et al., 2020) have been developed to enhance robustness of deep learning models against attacks, such as the preprocessing based methods (Liao et al., 2018; Jiao et al., 2023) and training provably robust networks (Dvijotham et al., 2018; Wong & Kolter, 2018). Among these defense methods, Adversarial Training (AT) (Madry et al., 2017) is widely recognized to be one of the most effective strategies. Based on the min-max formulation of Standard AT (SAT), single-step AT methods such as Fast-AT (Wong et al., 2020) are proposed to implement computation-efficient fast training. Fast-AT-GA (Andriushchenko & Flammarion, 2020) adopt implicit GA regularization which yields better

performance than Fast-AT. In recent works, Fast-BAT (Zhang et al., 2022) incorporates the Implicit Gradient (IG) to estimate the hyper gradient based on the Bilevel Optimization (BLO) formulation and obtains the state-of-the-art performance. In addition, PGD based AT methods (Pang et al., 2020) have been continuously improved by introducing heuristic techniques (Zhang et al., 2020; Carlini et al., 2022) and prior knowledge (Chen et al., 2020; Dong et al., 2022; Latorre et al., 2023) from other domains. In this paper, we focus on how these methods assist the target model itself react to the adversarial attacks based on the SAT process with online or offline computation cost, and propose a new proxy based adversarial defense framework to boost robustness.

## A.2 EXPERIMENTAL SETTING

**Datasets, models and metrics.** In this paper, we conduct our experiments based on CIFAR10 dataset (Krizhevsky et al., 2009) and the larger CIFAR100 dataset (Krizhevsky et al., 2009), which are commonly used for AT. As for the network structures, we mainly use the PreActResNet (PARN)-18 (He et al., 2016) as our backbone, and also implement the WideResNet (WRN)-34-10 (Zagoruyko & Komodakis, 2016) model to demonstrate the generalization performance of LAST framework with large-scale network structure. Two widely known adversarial attacks are selected for robustness evaluation, i.e., PGD and AutoAttack (Croce & Hein, 2020). Specifically, we use PGD-10 and PGD-50 to represent 10-step PGD attack with 1 restart step and 50-step PGD attack with 10 restart steps, respectively. We adopt the test Robust Accuracy (RA) as the metric of robust evaluation and report the average runtime for each iteration within the firs epoch to compare the computation cost of original AT methods and our improved ones. In particular, we also analyze the convergence behavior of test robust loss and RA to evaluate the robustness performance and stability of training process. Note that we report the average performance of the best model trained with early stopping for all the methods.

**Baselines and training details.** Since the proposed LAST adversarial defense framework essentially design new updates rule of defense, it has the potential to consistently improve various popular SAT methods. For the single-step AT methods, we choose Fast-AT (Wong et al., 2020), Fast-AT-GA (Andriushchenko & Flammarion, 2020) and Fast-BAT (Zhang et al., 2022) as representative SAT methods. Fast-BAT accesses the second-order gradient with Implicit Gradient (IG) based on BLO formulation, which represents the state-of-the-art baseline for comparison. As for the multi-step AT methods, PGD based AT have been widely explored and used for improving robustness with higher computational cost. Therefore, we choose these four methods to show that the proposed framework could consistently and universally enhance established SAT approaches. Basically, we follow (Zhang et al., 2022; Andriushchenko & Flammarion, 2020) to set most common hyperparameters such as the attack learning rate $\alpha$ for different adversarial attacks. For PARN-18, we train Fast-AT, Fast-AT-GA, Fast-BAT and PGD-AT(-2) (2-step PGD-AT) for 30 epochs and use cyclic scheduler with maximum $\beta = 0.2$, and train PGD-AT(-10) (10-step PGD-AT) for 200 epochs using multi-step scheduler with $\beta = 0.1$. For WRN-34-10, we train the single-step AT methods for 50 epochs using the same learning rate schedular with $\beta = 0.1$. As for the hyper parameter, we use same $\beta$ and $\alpha$ as the original AT methods, and set the aggregation coefficient $\gamma = 0.8$. When implementing Fast-AT and our version under perturbation $\epsilon = 16/255$ in Fig. 6, we set $\beta = 0.1$, $\gamma = 0.4$, $\mu = 0.95$ and $\tau = 6.0$.

## A.3 MORE RESULTS FOR ABLATION STUDY

**Ablation results with larger perturbation size.** In Fig. 6, we provide more detailed comparative results of the convergence behavior of test robust loss and RA to demonstrate the effectiveness of SD objective on CIFAR10 dataset with $\epsilon = 16/255$, which serves as a supplement to Fig. 3. As it is shown, combining the LAST framework will slow down the collapse of loss and accuracy and improve the best performance to some extent. When we integrate the LAST framework together with SD defense objective, the convergence behavior of test robust loss and RA are consistently improved and leads to boost of robustness. Unless specified otherwise, we only implement the LAST framework without SD objective to report the performance based on the same configuration of defense objectives and common hyperparameters.

**Ablation results on the influence of aggregation coefficient.** In Fig. 7, we further investigate the influence of hyperparameter unique to our method, the aggregation coefficient $\gamma$ based on the

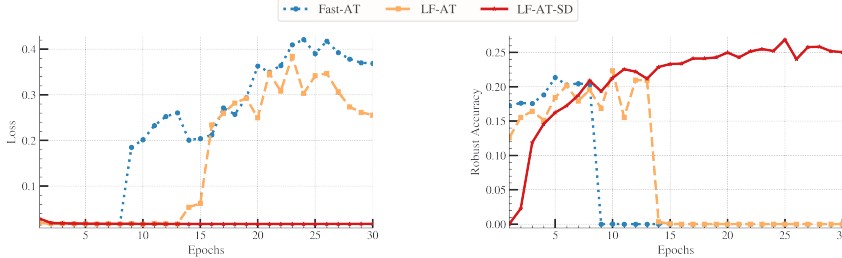

Figure 6: We compare the convergence behavior of Fast-AT, the improved version under LAST framework with and without SD defense objective (denoted as LF-AT and LF-AT-SD, respectively).

model trained on CIFAR10 dataset, $\epsilon = 8/255$. Specifically, we implement Fast-AT and our LAST framework without SD defense objective, and report the test robust loss and RA as $\gamma$ changes. Firstly, it can be observed that also the improved method converges a bit slower than Fast-AT at the beginning, it gains lower loss and higher RA as the training proceeds. Secondly, although our methods always gain better performance than Fast-AT as $\gamma$ changes, the RA slightly decreases when $\gamma$ approaches 1 ($0.8 \rightarrow 1.0$). Based on the above observation, we set $\gamma = 0.8$ for the above quantitative experiments unless specified otherwise.

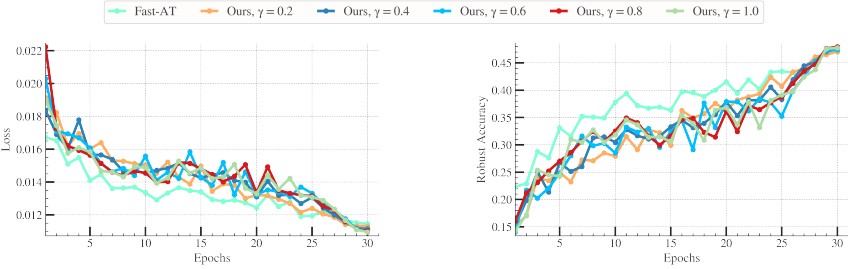

Figure 7: Illustration of convergence behavior of Fast-AT and our improved version under LAST framework without SD defense objective as $\gamma$ varies.

### A.4 COMPLETE EVALUATION RESULTS WITH LARGER MODELS AND DATASETS

Table 4: We report the SA and RA of Fast-AT, Fast-AT-GA, Fast-BAT and the LAST framework under PGD attack (PGD-10 and PGD-50) and AutoAttack on CIFAR10 dataset using WRN-34-10 structure as the backbone.

| CIFAR-10 dataset, WRN-34-10 trained with $\epsilon = 8/255$ | | | | | | |
|---|---|---|---|---|---|---|
| Method | SA (%) | PGD-10 (%) | | PGD-50 (%) | | AutoAttack (%) | |
| | | $\epsilon = 8/255$ | $\epsilon = 16/255$ | $\epsilon = 8/255$ | $\epsilon = 16/255$ | $\epsilon = 8/255$ | $\epsilon = 16/255$ |
| Fast-AT | **80.00** | 45.89 | 17.49 | 43.65 | 10.92 | 41.12 | 7.80 |
| Fast-AT-GA | 78.72 | 46.82 | 18.01 | 45.12 | 12.31 | 42.81 | 9.82 |
| Fast-BAT | 79.93 | 47.87 | 17.55 | 46.45 | 12.41 | 43.99 | 10.09 |
| LAST(Ours) | 77.88 | **49.02** | **19.23** | **47.94** | **14.15** | **45.49** | **11.87** |

**Results on the WRN-34-10 backbone.** To verify the generalization performance of LAST framework, we also report the RA and SA of the above single-step AT methods and the LAST framework on the larger network structure, i.e., WRN-34-10. Note that we implement our LAST framework

based on Fast-AT-GA, and run the above methods by setting the random seed as 1. It can be observed that our LAST framework also gains significant performance improvement on WRN-34-10 under $\epsilon = 8/255$ and $\epsilon = 16/255$.

Table 5: Illustrating the SA and RA of Fast-AT (F-AT), Fast-AT-GA (F-AT-GA) ,Fast-BAT (F-BAT) and corresponding improved version under PGD attack (PGD-10 and PGD-50) and AutoAttack on CIFAR100 dataset. We use $m\pm n$ to denote the mean SA (i.e., $m$) with standard deviation (i.e., $n$).

| CIFAR-100 dataset, PARN-18 trained with $\epsilon = 8/255$ | | | | | | |
|---|---|---|---|---|---|---|
| Method | SA (%) | PGD-10 (%) | | PGD-50 (%) | | AutoAttack (%) |
| | | $\epsilon = 8/255$ | $\epsilon = 16/255$ | $\epsilon = 8/255$ | $\epsilon = 16/255$ | $\epsilon = 16/255$ |
| F-AT | **55.087**±1.05 | 24.330±0.71 | 7.430±0.22 | 23.533±1.05 | 5.520±0.07 | 4.153±0.19 |
| LF-AT | 50.817±0.33 | **25.190**±0.07 | **9.003**±0.07 | **24.373**±0.42 | **7.497**±0.09 | **5.443**±0.13 |
| F-AT-GA | **53.253**±0.27 | 25.660±0.10 | 8.603±0.05 | 24.853±0.06 | 6.836±0.10 | 5.320±0.03 |
| LF-AT-GA | 48.220±0.32 | **25.887**±0.14 | **10.277**±0.10 | **25.433**±0.23 | **8.813**±0.14 | **6.270**±0.12 |
| F-BAT | **42.793**±8.42 | 22.603±3.85 | 8.920±0.67 | 22.059±3.24 | 7.813±0.34 | 5.807±0.60 |
| LF-BAT | 42.460 ±9.73 | **23.153**±5.76 | **9.840**±2.37 | **22.783**±5.56 | **8.740**±1.91 | **6.103** ±1.45 |

**Full results on CIFAR 10 and CIFAR100 datasets.** In addition to Tab. 2 and Tab. 3, we provide detailed comparative results with the corresponding standard deviation on CIFAR10 and CIFAR100 in Tab. 5 and Tab. 6. Due to space limit, we use F-AT to denote our method implemented based Fast-AT, and other methods use similar abbreviations. All the results are calculated by running the training process with three random seeds, since it is found that the performance of Fast-BAT will make a significant difference using its default seed and other random seeds when trained on the larger CIFAR100 dataset. As it has been verified before, both single-step and multistep AT methods improved by the LAST framework shows consistently better defense capability against representative adversarial attacks under different perturbation sizes.

Table 6: We report the SA and RA of PGD-10 and our improved version (LPGD-10) under PGD attack (PGD-10 and PGD-50) and AutoAttack on CIFAR10 and CIFAR100 dataset. We use $m\pm n$ to denote the mean SA (i.e., $m$) with standard deviation (i.e., $n$).

| CIFAR-10 dataset, PARN-18 trained with $\epsilon = 8/255$ | | | | | | |
|---|---|---|---|---|---|---|
| Method | SA (%) | PGD-10 (%) | | PGD-50 (%) | | AutoAttack (%) |
| | | $\epsilon = 8/255$ | $\epsilon = 16/255$ | $\epsilon = 8/255$ | $\epsilon = 16/255$ | $\epsilon = 16/255$ |
| PGD-AT | 81.948±0.74 | 51.923±0.30 | 20.310±0.75 | 50.757±0.33 | 15.677±0.41 | 13.093±0.43 |
| LPGD-AT | **82.187**±0.90 | **53.230**±0.20 | **22.203**±0.37 | **52.137**±0.10 | **17.587**±0.57 | **14.297**±0.03 |
| CIFAR-100 dataset, PARN-18 trained with $\epsilon = 8/255$ | | | | | | |
| Method | SA (%) | PGD-10 (%) | | PGD-50 (%) | | AutoAttack (%) |
| | | $\epsilon = 8/255$ | $\epsilon = 16/255$ | $\epsilon = 8/255$ | $\epsilon = 16/255$ | $\epsilon = 16/255$ |
| PGD-AT | **49.457**±0.48 | 25.837±0.43 | 9.980±0.43 | 25.377±0.39 | 8.749±0.38 | 6.667±0.23 |
| LPGD-AT | 48.150±0.42 | **31.267**±0.53 | **14.903**±0.26 | **30.857**±0.62 | **13.573**±0.52 | **8.033**±0.40 |

