# OpenReview forum: "Learn from the Past: A Proxy based Adversarial Defense Framework to Boost Robustness"
_ICLR.cc/2024/Conference — ICLR 2024 Conference Withdrawn Submission_

### Official Review · Reviewer_atuW · 2023-10-31

**Soundness:** 3 good
**Presentation:** 3 good
**Contribution:** 3 good
**Rating:** 6
**Confidence:** 4

**Summary:**

This paper proposes a framework called LAST which can be combined with existing adversarial training techniques to improve robustness without much increase in computational complexity.  The LAST framework utilizes a proxy model which stores a previous state of the target model that is being trained and uses this information when determining the parameter update of the model.  They empirically demonstrate that using this gradient update leads to smoother loss landscapes and improvements in robust performance.

**Strengths:**

- paper is well-written and ideas are clear
- good scope in experiments: authors experiment with multiple datasets (CIFAR-10, CIFAR-100) and combine LAST framework with multiple single step and multi-step AT training techniques to demonstrate improvements.  Also interesting experiments on transfer attacks between models.
- introduction of method is clear, authors give a good intuition for why LAST framework can help with robustness

**Weaknesses:**

- It would also be nice to see experiments with other model architectures and L2 threat model
- some improvements in robustness (especially combined with single step AT training) are very small (error bars overlap)

**Questions:**

See Weaknesses

---

> ### Author Response · Authors · 2023-11-13
> **To Reviewer #4**
>
> We thank the reviewer for positive comments and insightful questions. Our response to the reviewer’s question is below.
>
> **Q1.**  *``It would also be nice to see experiments with other model architectures and L2 threat model.''*
>
> **R1.** Thank you for acknowledging the innovation in our approach. We appreciate your suggestion, and in the supplementary material, we have conducted experiments using the WideResNet-34-10 architecture, which has a larger parameter count, and included the corresponding results. Regarding the L2 threat model, we recognize its importance, and we are actively working on conducting experiments to provide insights into the performance of our proposed defense framework under L2 attacks. Additionally, considering the input from other reviewers, we are committed to supplementing our results with experiments on different datasets. We hope these additions will address your concerns and contribute to a more comprehensive evaluation of our method.
>
>
> **Q2.** *``some improvements in robustness (especially combined with single step AT training) are very small (error bars overlap).''*
>
> **R2.** Thank you for your inquiry. We appreciate your observation regarding the small improvements in robustness, especially when combined with single-step AT. It is important to note that our intention was to provide a fair evaluation by conducting multiple trials with different random seeds. The observed overlap in error bars for some methods is indicative of their inherent instability and varying performance. In contrast, our proposed defense framework demonstrates not only lower standard deviation but also consistent performance improvement. The training curves and adversarial loss surfaces presented in the paper support the claim that our framework enhances training stability and defense against stronger, unknown perturbations. We thank you for recognizing the merits of our defense framework and hope that the additional experimental results we plan to include will further address your concerns.

---

> > ### Author Response · Authors · 2023-11-21
> > **Additional Results on Larger Dataset and Reminder--Discussion Stage 1 closing soon**
> >
> > Dear Reviewer atuW,
> >
> > We value your comments in rebuttal. As described in the reply at the beginning of rebuttal, since experiments with larger network architectures (i.e., WideResNet-34-10) have been shown in the Appendix, during rebuttal we trained Fast-AT with the L2 norm attack as well as improved versions under our proposed framework, based on different attack types and attack strengths are carried out. In addition, we also consider the suggestions given by different reviewers, and supplement the experimental results respectively (more results on different datasets, methods and out-of-distribution samples). These results are available in tabular form in the responses. We will also improve more complete results under different attack norms in subsequent versions, so as to more fully verify the performance of the proposed defense framework.
> >
> > | Methods | Clean|PGD-10 $\epsilon=8/255$  |PGD-10 $\epsilon=16/255$| PGD-50 $\epsilon=8/255$  |PGD-50 $\epsilon=16/255$| AutoAttack $\epsilon=8/255$|AutoAttack $\epsilon=16/255$|
> > | :---: | :----: | :----: | :----:  | :----: | :---: | :---: |  :---: |
> > | Fast-AT |$93.27$| $90.86$ | $87.81$ | $90.73$ | $87.54$ | $90.68$|$87.43$ |
> > | Ours | $93.49$ | $\mathbf{91.26}$ | $\mathbf{88.59} $| $\mathbf{91.16} $ | $\mathbf{88.33}$ |$\mathbf{91.08}$ | $\mathbf{88.27}$ |
> >
> > We appreciate the time and effort that you have dedicated to reviewing our manuscript. Just a quick reminder that discussion stage 1 is closing soon.
> >
> > Have our responses addressed your major concerns?
> >
> > If there is anything unclear, we will address it further. We look forward to your feedback.
> >
> > Best, Authors of Paper 1779

---

### Official Review · Reviewer_MeBp · 2023-11-08

**Soundness:** 2 fair
**Presentation:** 2 fair
**Contribution:** 2 fair
**Rating:** 3
**Confidence:** 5

**Summary:**

The paper proposes LAST, a framework for training adversarially robust classifiers. At each optimization step, the adversarial perturbations are computed on the target model, but, unlike in commonly used methods, the model parameters update is computed via a proxy model, which can be seen as a past version of the target model. Such framework has the advantage of reducing the instabilities of robust training like catastrophic overfitting, and can be used in combination with different popular variants of adversarial training. In the experimental evaluation on CIFAR-10 and CIFAR-100, LAST provides improvements in the adversarial robustness of the classifiers.

**Strengths:**

- The approach of using a proxy model for computing the updates for stabilizing adversarial training is novel.

- The models trained with LAST achieve higher robustness than the baselines at high perturbation radii.

**Weaknesses:**

- The presentation is not clear: there are non-standard expressions like "parameter-oriented attack" which are not defined, and sentences difficult to follow and interpret (e.g. first paragraph in Page 4).

- The motivation behind the proposed algorithm, and the steps followed to formulate it, is not clear. If I'm not missing something, I think that LAST can be summarized as computing the adversarial attack on the current model, transferring it to the model at the previous iteration, computing a gradient step on it, and updating the current model with such step (plus SWA). This has the effect of slowing down training until the current and previous model are similar, i.e. end of training with small learning rate, and prevent (or delay) overfitting. However, it is not clear why this should provide higher robustness in general. Additionally, the SD loss looks like a variant of the TRADES loss (Zhang et al., 2019) where cross-entropy is used on the adversarial instead of clean points.

- In Table 1, the difference in robustness at $\epsilon=8/255$, which is the target threat model seen during training, evaluated with AutoAttack, between the models with LAST and baselines is within the standard deviation, while LAST models have significantly worse clean performance. In Table 2 and Table 3, the robust accuracy at $\epsilon=8/255$ with AutoAttack is not reported. Then, the improvement provided by LAST in such threat model is not evident.

- For the comparison with multi-step methods, there should be a comparison with SOTA techniques (see e.g. [A]) to support the benefit provided by LAST in such case.

[A] https://robustbench.github.io/

**Questions:**

See details above.

Overall, I think the experiments suggest that LAST provides some improvement in robust accuracy when combined single-step methods and evaluated at larger radii than what used for training (which is not a common evaluation setup), but at the cost of worse standard accuracy. This should be more clearly conveyed when stating the contributions of the paper. Moreover, the presentation and motivation for the proposed method need improvement in my opinion.

---

> ### Author Response · Authors · 2023-11-13
> **To Reviewer #3**
>
> Thank you for insightful comments and interesting suggestions to improve our work. We also appreciate the amount of your time on reviewing our paper. We address your concern individually.
>
> **Q1.**  *``The presentation is not clear: there are non-standard expressions like "parameter-oriented attack" which are not defined, and sentences difficult to follow and interpret (e.g. first paragraph in Page 4).''*
>
> **R1.** Thank you for your feedback. We appreciate your observation regarding non-standard expressions, such as "parameter-oriented attack," and acknowledge the importance of providing clear definitions for key terms from the outset.
> 1. Indeed, we introduced the term "parameter-oriented attack" in *Section 2.2*, where we stated, "Since the perturbations generated in these attacks are always targeted at the current state of the model parameters, they can be considered attacks oriented towards the parameters." This term encapsulates a crucial characteristic derived from our observations of transfer attacks and white-box attacks, forming a key motivation for introducing historical states to aid in defense.
> 2. Regarding the passage on the first paragraph of *Page 4*, we aimed to elucidate the close relationship between generated adversarial attacks and the current parameter state. However, due to label noise and the complex mapping between parameter gradients and attacks introduced by operations like "sign," relying solely on the current parameter state for defense may not be the optimal choice.
>
>
> We value your input on improving the clarity of this core expression. In future revisions, we will provide explicit explanations and employ more concise language to articulate our motivations.
>
>
> **Q2.**  *``The motivation behind the proposed algorithm, and the steps followed to formulate it, is not clear. If I'm not missing something...''*
>
> **R2.**  Thank you for your inquiry, and your questions regarding the effectiveness of the proposed defense framework and its similarities to existing techniques are valuable. Here are clarifications on the motivation, core methodology, and the relation to existing methods within the proposed defense framework:
>
>
> 1. *Advantages of the Proxy Model in Defense:* The defense against adversarial attacks generated based on the current model state, leveraging the historical state of model parameters (proxy model), offers at least two advantages. Firstly, since these attacks are not tailored to the historical state, the proxy model provides effective priors to assist in the defense of the target model. Secondly, in this update scheme, we effectively utilize the proxy model to provide a better initial point for the defense process of the target model, employing an aggregation of parameters to conduct the subsequent attack.
> 2.  *Core Methodology in the Defense Framework:* Consequently, the core of this defense framework involves using the proxy model to defend against attacks based on the target model, with an update rule that transforms similarly to the SWA technique but with distinct emphasis. We provided an analysis of this in the discussion section.
> 3. *Performance of the Defense Framework:* This defense framework can achieve better defense performance based on the same number of training iterations and identical hyperparameters as the original algorithms. It converges to a state where the historical and current states are similar, demonstrating effectiveness in experiments based on PGD-AT, where our improved LPGD-AT exhibits more stable and smaller fluctuations in the final accuracy, as shown in *Figure 4*.
> 4. *Relation to Existing Techniques within the Framework:* You mentioned the similarity between the proposed SD loss and the TRADES loss function. However, in reality, the update form, especially with regards to temperature and distillation coefficients, is closer to the standard knowledge distillation loss. Moreover, the structure of this loss function is based on the introduced proxy model, representing an additional technique within our newly proposed defense framework.
>
>
> In conclusion, we appreciate your questions on the algorithm's structure and the introduced loss function within the defense framework. If you have further inquiries, we welcome continued discussion.

---

> ### Author Response · Authors · 2023-11-13
> **More Comments To Reviewer #3**
>
> **Q3.**  *``In Table 1, the difference in robustness at $\epsilon=8/255$, which is the target threat model seen during training, evaluated with AutoAttack, between the models with LAST and baselines is within the standard deviation, while LAST models have significantly worse clean performance. In Table 2 and Table 3, the robust accuracy at $\epsilon=8/255$ with AutoAttack is not reported. Then, the improvement provided by LAST in such threat model is not evident.''*
>
> **R3.** Thank you for your detailed scrutiny of the experimental results. I will further elucidate the issues raised regarding the experimental performance in academic English:
>
> 1. *Standard Deviation and Consistency Improvement:* Concerning $\epsilon=8/255$, we conducted a retraining of all algorithms, considering the impact of random seeds. Each algorithm underwent three independent training runs. Notably, the original methods exhibited a higher standard deviation, indicative of training instability. Conversely, our method demonstrated a lower standard deviation at $\epsilon=8/255$, underscoring a consistent performance enhancement.
> 2. *Performance at $\epsilon=16/255$ Enhancement:* Regarding $\epsilon=16/255$, we presented a significantly improved defense performance. This result validates the enhancements in our framework concerning the smoothness of adversarial loss surfaces and defense against stronger unknown attacks. It is noteworthy that the presentation of adversarial loss surfaces and training curves serves as a key means to highlight improvements over standard adversarial training.
> 3. *Reason for Missing Results at $\epsilon=8/255$:* We appreciate your observation of the absence of results for $\epsilon=8/255$ in the table. This omission is due to space constraints, prompting us to initially showcase results for $\epsilon=16/255$. In this response, we will first supplement the $\epsilon=8/255$ partial results on PGD-AT which have been evaluated before. Furthermore, in subsequent versions, we will comprehensively present all experimental outcomes.
>
> - Regarding the experimental results on AutoAttack in *Table 3* of the manuscript, the supplementary information is provided below:
> |PreActResNet-18 | CIFAR10 dataset | AutoAttack  | PreActResNet-18 | CIFAR100 dataset | AutoAttack  |
> | :---: | :--------: | :---: | :---: | :--------: | :---: |
> |  Method  | $\boldsymbol{\epsilon}=8 / 255$ | $\boldsymbol{\epsilon}=16 / 255$ | Method  | $\boldsymbol{\epsilon}=8 / 255$ | $\boldsymbol{\epsilon}=16 / 255$ |
> | PGD-AT | $47.087 \pm 0.54$ | $13.093 \pm 0.43$ | PGD-AT | $21.109 \pm 0.05$ | $6.667 \pm 0.23$ |
> | LPGD-AT (Ours) | $\mathbf{47.707}\pm 0.22$ | $\mathbf{14.297} \pm 0.03$ | LPGD-AT (Ours) | $\mathbf{23.113} \pm 0.18$ | $\mathbf{8.033} \pm 0.40$ |
>
> If you have any further inquiries regarding other experimental results, we welcome further questions for in-depth discussion.
>
> **Q4.**  *``For the comparison with multi-step methods, there should be a comparison with SOTA techniques (see e.g. [A]) to support the benefit provided by LAST in such case.''*
>
> **R4.** Thank you for raising the question regarding the comparison of multi-step methods. In our forthcoming revision, we plan to include benchmark SOTA techniques, as cited in reference [A], to provide a more comprehensive context for evaluating the efficacy of LAST in the domain of multi-step adversarial training.
>
> 1. We have previously outlined (in the second paragraph of *Appendix A.2*) the criteria for selecting the comparative methods in the paper. We believe that improvements observed on these methods can demonstrate the effectiveness of our proposed defense framework in enhancing training stability and robustness against various attack intensities, as indicated by the results on various training curves and surfaces.
> 2. Additionally, we highly value your suggestion on the SOTA methods for enhancing multi-step adversarial training. Therefore, we will make every effort to present experimental results based on methods [1] from RobustBench by the end of the rebuttal period and supplement more extensive performance results in subsequent versions.
>
>
> We sincerely appreciate your continued effort in reviewing our paper and providing valuable suggestions. We highly value your expertise and insights, and we genuinely look forward to further engaging with you to ensure the thorough evaluation of our paper. If there are any additional questions or if you require further clarification on specific aspects, we are more than willing to provide the necessary information.
>
> [1] Jia X, Zhang Y, Wu B, et al. LAS-AT: adversarial training with learnable attack strategy. CVPR 2022.

---

> > ### Author Response · Authors · 2023-11-21
> > **Addtional Results on Multi-step AT methods and Reminder--Discussion Stage 1 closing soon**
> >
> > Dear Reviewer MeBp,
> >
> > We take your comments in rebuttal very much into account, and in this subsequent period, we have supplemented our latest results based on our new multi-step attack method LAS-AT[1] (from RobustBench) on the TinyImageNet dataset, tested on different attacks. Hopefully, the new results address the benefit on SOTA methods that you were concerned about. We will consider mode different techniques and supplement more detailed results in the subsequent versions to more fully verify the performance of the proposed defense framework.
> >
> > | Methods | Clean |PGD-10   | PGD-20 | PGD-50  | AutoAttack|
> > | :---: | :----: | :----: | :----:  | :----: | :---: |
> > | LAS-AT |$40.48$| $17.53$ | $17.26$ | $17.17$ | $13.05$ |
> > | L-LAS-AT (Ours) | $38.54$ | $\mathbf{19.95}$ | $\mathbf{19.73} $| $\mathbf{19.67} $ | $\mathbf{14.33}$ |
> >
> > We appreciate the time and effort that you have dedicated to reviewing our manuscript. Just a quick reminder that discussion stage 1 is closing soon.
> >
> > Have our responses addressed your major concerns?
> >
> > If there is anything unclear, we will address it further. We look forward to your feedback.
> >
> > Best, Authors of Paper 1779

---

> > > ### Comment · Reviewer_MeBp · 2023-11-21
> > >
> > > I thank the authors for the response and additional experiments.
> > >
> > > I still see most of the weaknesses mentioned in the original review: using the model from the previous iteration to compute the update step doesn't seem to have a concrete motivation, especially for achieving higher robustness. Moreover, the form of the presentation (text clarity) would need improvement in my opinion.
> > >
> > > About the experimental results, if the goal is to improve single-step AT methods, I think the effectiveness of LAST is not clear (see comment in the initial review): a significant amount of clean performance is lost for little or no improvement of robustness in the target threat model (8/255), and some gain at the larger radius which is however not the main objective (Table 1 and Table 2). For multi-step methods, I think a more comprehensive comparison to existing methods is needed: I appreciate that the authors further tested the effect of the proposed method in combination with LAS, but there are several methods achieving more robust classifiers on CIFAR-10 (see RobustBench).
> > >
> > > Overall, I think the proposed method is currently not sufficiently supported by a clear motivation or experimental evidence.

---

> ### Author Response · Authors · 2023-11-21
> **Further comments**
>
> We sincerely appreciate the prompt feedback from the reviewer. In addressing the noted weaknesses and the raised queries concerning additional experimental results, we provide detailed clarifications as follows:
> 1. **Clarification of Terminology**: Regarding the term "parameter-oriented" attack mentioned in your feedback, we have now provided a more detailed explanation. Notably, this concept is also discussed before in the manuscript.
> 2. **Algorithmic Motivation**: Concerning the motivation behind our proposed algorithm, we acknowledge your feedback. Fig. 2 is presented as a clear visual explanation. The historical state of model updates during adversarial training contributes to a more robust defense against gradient-based attacks. This serves as a form of prior knowledge aiding the model's current updates. Additionally, distinctions from existing techniques are elaborated in the discussion section. We also appreciate your recognition of the novelty of employing a proxy model in the adversarial defense process, and we will consider more experimental results about this tradeoff phenomenon.
> 3. **Performance Improvement under Training Attack Strength**: The observed variance in the performance of standard adversarial training has been acknowledged. Our method consistently enhances the average performance of existing methods while exhibiting a smaller standard deviation, demonstrating superior generalization. Besides, the robustness across different attack strengths is evident from the adversarial loss surface analysis, highlighting the practicality of our defense framework in real-world scenarios. Because in real scenarios, it is often not guaranteed that the strength of adversarial attacks is exactly the same as the strength of training.
> 4. **Comparison with Multi-step Adversarial Training**: Acknowledging time constraints, we expanded our experimentation to a larger dataset, as suggested by another reviewer. Consequently, we conducted comparative experiments on the LAS-AT method on TinyImageNet. We are open to incorporating additional results focusing on the improvement of a single dataset in future revisions.
> 5. **Performance Trade-off on Clean Datasets**: Acknowledging the trade-off between model accuracy on clean datasets and adversarial robustness, we emphasize that this is an inherent challenge in adversarial training. Furthermore, existing techniques such as reducing label noise and leveraging class activation mapping can be applied to optimize this trade-off. The strong generalization of our framework allows the incorporation of such techniques to address this challenge.
> 6. **Experimental Design and Generalization**: We emphasize that our experimental design prioritizes assessing the generalization of the proposed framework on widely accepted SAT methods rather than aiming for dataset-specific optimizations. Both training curves and adversarial loss surfaces affirm the effectiveness of our defense framework.
>
> Should you have any further inquiries, we welcome continued discussion. Finally, we hope that, considering our responses, you may reconsider your evaluation of the manuscript.

---

### Official Review · Reviewer_9nzt · 2023-11-08

**Soundness:** 2 fair
**Presentation:** 3 good
**Contribution:** 1 poor
**Rating:** 3
**Confidence:** 4

**Summary:**

The paper presents a self-distillation based adversarial training framework that learns from past states to make models adversarially robust. In particular, the paper proposes a two-state weight update rule using the gradient of the past state to the current adversarial perturbation. Results on CIFAR-10 and CIFAR-100 show promise.

**Strengths:**

+ The idea of using past model states to improve the model’s defense against adversarial attacks is interesting.
+ Paper is in general easy to follow.

**Weaknesses:**

**Formulation**
* Considering the proposed method is a two-phase approach, what is its implication on the training time/computational complexity of the method, when compared to baselines? This may be very important to address, since the paper by itself refers to SAT as a computationally expensive approach. How does the proposed method compare on this?
* The paper lacks a deeper analysis of the proposed method. For e.g., in Algorithm 1, substituting L13 in L16 yields \tilde{w} as an additive component to \theta. This would imply gradient ascent in the direction updated in L12. What does this imply?
* Why does the model state from the past expected to behave more robustly to perturbations? While an empirical motivation is provided, a more intuitive (or conceptual or theoretical) motivation is required.
* Sec 2.4 discusses briefly how the momentum may be different from the use of momentum. It would have been better to see empirical results with just momentum to indeed show that this is the case.

**Experiments**
* It is not clear how the baseline methods for picked for the experimental studies. There is no discussion or motivation for the choices. Compared to recent papers on adversarial training, the number of baselines studied seem fewer.
* All through the methodology section (Sec 2), there was no mention that the proposed method is an add-on to existing methods. The experiments however only study the method as an add-on, not as a separate stand-alone method. Why is this the case? Without a clear reason for this decision, this makes the contributions weak.
* Continuing with the previous point, how are the methods in the experiments – LF-AT, LF-AT-GA, LF-BAT, etc - implemented? Without this detail, it is difficult to follow the results.
* Results on CIFAR-10 and CIFAR-100 may not be conclusive – it is important to study datasets of the scale of ImageNet.
* Why was PARN-18 chosen as the backbone? Why not just ResNet-18 or any other backbone?
* In Table 2, using the \uparrow to show improvement across 3 random seeds seemed rather unconventional. Generally, papers report mean and std dev when there are multiple trials. It was not clear what was being conveyed here.

**Literature**
* The discussion of related work is limited and weak. Beyond not being comprehensive for adversarial attacks and defenses, the paper also misses discussing other earlier efforts that look back at past model states. E.g  Jandial et al, Retrospective Loss: Looking Back to Improve Training of Deep Neural Networks, KDD 2020 (Although this work did not explicitly look at adversarial robustness, it is important to discuss literature comprehensively)

**Clarity and Presentation**
* I found the plots in Fig 1 difficult to understand – what is loss gap? Between what quantities?

**Questions:**

Please see weaknesses above

---

> ### Author Response · Authors · 2023-11-13
> **To Reviewer #2 About Formulation Part 1**
>
> We thank the reviewer for recognizing the originality and significance of our work. Thank you for your in-depth technical considerations regarding our manuscript. Your suggestions are highly valuable for our improvement. We address your concern individually.
>
>
> **Q1.**  *``Considering the proposed method is a two-phase approach, what is its implication on the training time/computational complexity of the method, when compared to baselines? This may be very important to address, since the paper by itself refers to SAT as a computationally expensive approach. How does the proposed method compare on this?''*
>
> **R1.**  We appreciate the reviewer's thoughtful inquiry regarding the computational burden of our proposed method. Addressing this concern aligns with our recognition that the computational efficiency of our defense framework is pivotal for its practical applicability, especially considering the computational expense associated with Standard Adversarial Training (SAT) mentioned in the manuscript.
>
> 1. In *Table 1*, we present the runtime for three adversarial training algorithms, both before and after improvement. It is evident that the increase in runtime is negligible (only $1 \times10^{-3} -2 \times10^{-3} $ sec/iteration), as reported. This experimental outcome aligns with the description in *Algorithm 1*, where the primary computational cost of our proposed proxy-based defense framework occurs in *Line 12*, involving the update of the proxy model. This computation cost is analogous to the computational overhead incurred by SAT framework during the defense process. The updates in other parts of the process involve rapid parameter update, incurring minimal computational burden.
> 2. The central theme of our discourse in this manuscript is the realization that many enhancements to adversarial training methods often introduce substantial computational overhead, whether during the training process or as a consequence of pre-training. This underscores our particular emphasis on single-step adversarial training methods, chosen for their expeditious and efficacious defensive strategies when compared to the more computationally demanding nature of standard adversarial training. Notably, we have improved three single-step adversarial training algorithms within our framework with only slight increase of runtime, demonstrating consistent performance enhancements as illustrated in the *first paragraph of Section 3.1*.
>
>
> **Q2.**  *``The paper lacks a deeper analysis of the proposed method. For e.g., in Algorithm 1, substituting L13 in L16 yields $\tilde{w}$ as an additive component to $\theta$. This would imply gradient ascent in the direction updated in L12. What does this imply?''*
>
> **R2.** Thank you for your insightful comments. The issue you raised is indeed crucial and has been extensively addressed in *Section 2.4* of the manuscript (*2nd and 3st paragraph in Page 6*).
>
> 1. Initially, it is essential to note that we did not explicitly express the algorithm in the form you described. This intentional choice was made to facilitate a more nuanced analysis of the primary operations conducted in each of the two stages. The first stage involves preparing the gradients required to update the target model $\theta$, denoted as $\mathcal{G}_{{\theta}}$, while the second stage focuses on executing the actual update of the target model.
> 2. As detailed in the analysis presented in *Section 2.4* and consistent with your described operation, the substitution effectively aggregates the weights of the updated proxy model and the target model, i.e., $\theta_{i+1}=(1-\gamma) \cdot \theta_{i} + \gamma \cdot  \tilde{\omega}$. Leveraging this aggregation operation for updating the target model introduces a historical state to defend against adversarial attacks generated based on the current target model.
> 3. From this perspective, the additive part essentially acts as an effective form of prior knowledge, a unique contribution not present in prior methods. Besides, the mechanism of this defense framework are also inspired by similar ideas in commonly used momentum algorithms. The merits of this approach different from these momentum techniques are further discussed in the analysis section.
>
>
> We hope this clarification addresses your concerns adequately. Your feedback is invaluable, and we are committed to incorporating these details more explicitly in the revised manuscript.

---

> ### Author Response · Authors · 2023-11-13
> **To Reviewer #2 About Formulation Part 2**
>
> **Q3.**  *``Why does the model state from the past expected to behave more robustly to perturbations? While an empirical motivation is provided, a more intuitive (or conceptual or theoretical) motivation is required.''*
>
> **R3.** Thank you for your insightful question; it is indeed pivotal and encapsulates the core motivation behind our algorithm design.
>
> 1. To commence, our inspiration is drawn from transfer-based black-box attacks, as expounded in the second paragraph of *Section 2.2*. As highlighted, attacks generated based on specific model parameters exhibit diminished efficacy when applied to other black-box models. In the context of adversarial training, attacks on the current target model, rooted in its current parameter state, manifest reduced effectiveness when applied to the model's past parameter states. Therefore, integrating the historical state of model parameters as a defense mechanism constitutes a cost-effective and efficacious prior, aiding in the augmentation of the current model.
>
> 2. Additionally, the discussion part mentioned several instances where leveraging a model's prior states has assisted in updating the current model across diverse perspectives, as elaborated in *Section 2.4*. We emphasize that the update of proxy model actually incorporate prior information from historical states to help update the current parameters of target model, which have not been explored before.
>
> While we have addressed the fundamental aspects of your inquiry and the motivation behind our paper, we acknowledge that the content might benefit from greater focus and depth.
>
> **Q4.**  *``Sec 2.4 discusses briefly how the momentum may be different from the use of momentum. It would have been better to see empirical results with just momentum to indeed show that this is the case.''*
>
> **R4.** Thank you for your thoughtful inquiry. We appreciate your attention to the discussion on momentum in *Section 2.4*. In the context of SAT algorithms, including the four methods used for comparison in the paper, existing approaches always incorporate momentum-based update strategies when reporting their optimal performances. This inclusion is evident in the code provided in our supplementary material.
>
> Thus, in the empirical results we have presented, particularly in the comparison with methods like Fast-AT and PGD-AT, we have effectively demonstrated the superior effectiveness of our new update strategy compared to momentum. This aligns with the comparative experimentation you are interested in. Notably, detailed explanations regarding the use of optimizers based on momentum in our experiments are provided in *Section A.3* of the supplementary material. We acknowledge the importance of this analysis and will incorporate it explicitly into the documentation of our experimental results.
>
> Thank you for bringing up this point, and we are committed to addressing it comprehensively in the revised manuscript.

---

> ### Author Response · Authors · 2023-11-13
> **To Reviewer #2 About Experimental Part 1**
>
> **Q1.**  *``It is not clear how the baseline methods for picked for the experimental studies. There is no discussion or motivation for the choices. Compared to recent papers on adversarial training, the number of baselines studied seem fewer.''*
>
> **R1.** Thank you for your insightful question. The rationale for selecting the comparative methods is indeed discussed in the second paragraph of *Section A.2* in the supplementary material. We recognize the importance of this aspect and will provide further analysis in the subsequent revised version.
>
> 1. As previously mentioned, due to the practical utility of single-step adversarial training schemes, we have focused on enhancing several single-step methods in this paper, including Fast-AT, Fast-AT-GA, and Fast-BAT. Fast-BAT adopts a Bilevel optimization perspective for modeling adversarial training problems, offering more effective solution strategies. Given its relative complexity, it serves as a benchmark and a means to assess the consistent improvement of existing methods by our proposed framework.
>
> 2. PGD-AT is currently one of the most widely used multi-step adversarial training algorithms and has undergone numerous improvements. Therefore, we validated the effectiveness of our framework in the context of multi-step adversarial training algorithms based on PGD-AT.
> 3. Combining the above points, we have implemented a total of two categories and four methods. The results of these methods and the versions improved under our framework are all reported, respectively. This approach, in contrast to solely improving on the best-performing method and reporting state-of-the-art performance, provides a more comprehensive evaluation of the efficacy of our framework.
>
>
> We will thoroughly consider your suggestion regarding the analysis of the comparative methods and incorporate it into our subsequent versions.
>
>
> **Q2.**  *``All through the methodology section (Sec 2), there was no mention that the proposed method is an add-on to existing methods. The experiments however only study the method as an add-on, not as a separate stand-alone method. Why is this the case? Without a clear reason for this decision, this makes the contributions weak.''*
>
> **R2.** Thank you for your question; it raises a pertinent aspect of our methodology. Indeed, the description of our proposed method as an add-on is a valid characterization, and we appreciate your observation. However, throughout the paper, we consistently employ the term "framework" to describe our approach. The rationale for this choice is multifold:
>
> 1. We posit that most existing SAT methods can be re-implemented within our framework, allowing for a consistent enhancement of their defenses based on standard adversarial training.
> 2. The introduced SD defense loss function serves as an additional component within this new framework. This loss function, rooted in the introduced proxy model, cannot be directly applied to SAT methods. Thus, in the paper, we inted to use the terms "framework" and "add-on" to denote our proposed update strategy and the corresponding loss function.
> 3. Consequently, our experimental section primarily focuses on validating the effectiveness of various existing methods within our framework. Alternatively, one could view the improved versions of Fast-AT and PGD-AT, namely LFast-AT and LPGD-AT, as the most fundamental implementation of our framework.
>
>
>  If you find our explanation insufficient or have further queries, we welcome continued discussion to refine our analysis and the description of our framework.
>
>
> **Q3.**  *``Continuing with the previous point, how are the methods in the experiments – LF-AT, LF-AT-GA, LF-BAT, etc - implemented? Without this detail, it is difficult to follow the results.''*
>
> **R3.** Thank you for your insightful feedback, which underscores the crucial details regarding the enhancement of existing methods within our framework. In the second paragraph of *Appendix A.2*, we provided a concise elucidation and plan to furnish a more comprehensive explanation in subsequent revisions.
> 1. Our novel defense framework primarily focuses on fortifying the defense process, thereby accommodating both single-step and multi-step adversarial training methods.
> 2.  As explicated in the second paragraph of *Appendix A.2*, we strategically embedded the a priori information of the proxy model into the defense processes of various algorithms based on the code repository of Fast-BAT [A]. Concurrently, we maintained consistency with the original algorithms in terms of most hyperparameters except for parameters unique to the framework of our algorithm, such as the learning rate $\gamma$.
>
> Your feedback is invaluable, and we welcome further discussion or additional queries to ensure the clarity and coherence of our manuscript.
>
> [A] Zhang Y, Zhang G, Khanduri P, et al. Revisiting and advancing fast adversarial training through the lens of bi-level optimization. ICML, 2022.

---

> ### Author Response · Authors · 2023-11-13
> **To Reviewer #2 About Experimental Part 2**
>
> **Q4.**  *``Results on CIFAR-10 and CIFAR-100 may not be conclusive – it is important to study datasets of the scale of ImageNet. Why was PARN-18 chosen as the backbone? Why not just ResNet-18 or any other backbone?''*
>
> **R4.**  *Thank you for raising pertinent inquiries concerning the choice of datasets and model architecture, both of which we address collectively below:*
>
> 1. We highly value your suggestion regarding experiments on larger-scale datasets. Given the constraints on discussion time with the reviewer, we commit to diligently supplementing experimental results on the *TinyImageNet* dataset by the rebuttal deadline and further enriching subsequent versions with outcomes from additional datasets.
> 2. The prevailing practice in existing research involves the utilization of the *CIFAR-10 and CIFAR-100* datasets for assessing defense performance. Consequently, our selection of these datasets for evaluating the proposed framework's defense efficacy against diverse adversarial attacks is deemed sufficient to substantiate the framework's effectiveness.
> 3. The choice of *PreActResNet-18* as our experimental backbone stems from its status as an improved iteration of *ResNet-18*, exhibiting superior performance. Notably, it represents a commonly employed model for adversarial training in contemporary literature.
> 4. Furthermore, in the supplementary materials, we present corroborating results on the *WideResNet-34-10* model, characterized by an increased parameter count and commensurately enhanced defense efficacy.
>
> We sincerely appreciate your inquiries regarding the experimental aspects. Simultaneously, we are committed to promptly incorporating new analyses and experimental findings into the supplementary materials. Should you have any additional queries, we welcome your continued engagement for further discussion.
>
>
> **Q5.**  *``In Table 2, using the $\uparrow$ to show improvement across 3 random seeds seemed rather unconventional. Generally, papers report mean and std dev when there are multiple trials. It was not clear what was being conveyed here.''*
>
> **R5.** We appreciate your inquiry regarding the unconventional use of the $ \uparrow$ symbol in *Table 2* to denote improvement across three random seeds. Our rationale for adopting this presentation format is rooted in the desire to mitigate the potential impact of random seed variations on experimental outcomes. The specific considerations are as follows:
>
> 1. Through our experimentation, we observed deviations in experimental results when replicating these SAT methods using the original paper's provided random seed versus employing alternative random seeds. These deviations sometimes manifested as significant biases and, in extreme cases, led to catastrophic overfitting.
> 2. Consequently, to address this variability stemming from random seed fluctuations, we opted to retrain each algorithm three times using three different random seeds. We then computed the mean and standard deviation of the experimental results, a practice that aids in minimizing the influence of extraneous factors.
> 3. It is worth noting that the chosen presentation format aligns with established practices in prior works[A], where similar approaches have been employed. We contend that this methodology represents a reasonable and justifiable means of result presentation.
> 4. Furthermore, in the supplementary materials, we present corroborating results on the *WideResNet-34-10* model, characterized by an increased parameter count and commensurately enhanced defense efficacy.
>
>
> We trust that these clarifications shed light on the rationale behind our choice of presentation format. Should you have any further inquiries or suggestions, we welcome ongoing discussion on this matter.

---

> ### Author Response · Authors · 2023-11-13
> **To Reviewer #2 About Literature and Clarity Part**
>
> **Literature Part**
>
> **Q1.**  *``It is not clear how the baseline methods for picked for the experimental studies. There is no discussion or motivation for the choices. Compared to recent papers on adversarial training, the number of baselines studied seem fewer.''*
>
> **R1.** We appreciate your feedback on the references, as it holds significant importance for our study. We provide a detailed response to the raised concerns:
>
> 1. Acknowledging the limited discussion of existing adversarial attack and defense works in the current manuscript, we commit to expanding and enhancing this aspect in subsequent versions. The intention is to offer a more comprehensive overview of the relevant literature mentioned in the introduction.
> 2. Regarding the literature you referenced, upon further review, we acknowledge its applicability in diverse domains and its insightful implications for our research. We firmly assert that delving into the historical states of model parameters and enhancing their consistency holds paramount significance in the context of adversarial training. This is particularly crucial when considering the inherent characteristics of adversarial attacks targeting model parameters and the instability prevalent in the training process. Therefore, a thorough investigation into the historical states of model parameters stands as a crucial avenue for improving model robustness and adversarial resilience.
>
> We appreciate your guidance, and will augment the literature review section to present a more thorough discussion of related works in the revised version. Your valuable insights have been duly noted, and we welcome further discussions or suggestions you may have.
>
>
> **Clarity and Presentation Part.**
>
> **Q1.**  *``I found the plots in Fig 1 difficult to understand – what is loss gap? Between what quantities?''*
>
> **R1.** Thank you for your inquiry. We plot *Figure 1* to illustrate that the proposed defense framework is able to have better defensiveness against both adversarial and random perturbations at different intensities:
>
> 1. In the same coordinate system, we present the original and improved adversarial loss planes for four methods. By incorporating gradient-based adversarial perturbations and random perturbations into the plots, we analyze the model's robustness to perturbed images. This visual representation serves to intuitively demonstrate the effectiveness of the proposed framework.
> 2. Theoretically, models with stronger defense capabilities typically exhibit relatively smooth adversarial loss landscapes, with more pronounced fluctuations occurring only in the vicinity of the origin.
> 3. Additionally, the colorbars on the right side display the adversarial loss intervals for both the original and improved methods within the illustrated region. The difference between the maximum and minimum losses is also indicated, and denoted as the "loss gap". It can be observed that the methods under the new framework exhibit lower and smoother overall adversarial losses, with a smaller range. This indicates improved defense against both adversarial and random perturbations.
>
>
> We hope that the above explanation has helped address the concerns you have raised in your review. If there are other concerns or if you have more questions, we will be more than happy to provide additional clarification.

---

> > ### Author Response · Authors · 2023-11-21
> > **Additional Results on Larger Dataset and Reminder--Discussion Stage 1 closing soon**
> >
> > Dear Reviewer 9nzt,
> >
> > Previously, we tried our best to answer your related questions and suggestions for improvement at the beginning of the rebuttal. In this subsequent period, we supplemented the recent results on the TinyImageNet dataset based on the new multi-step attack method LAS-AT[B], tested on different attacks. Hopefully, the new results address the performance gains on larger datasets that you were concerned about in the experiments section. We also attach great importance to your comments and will supplement the results on more datasets in the subsequent versions to more fully verify the performance of the proposed defense framework.
> >
> > | Methods | Clean |PGD-10   | PGD-20 | PGD-50  | AutoAttack|
> > | :---: | :----: | :----: | :----:  | :----: | :---: |
> > | LAS-AT |$40.48$| $17.53$ | $17.26$ | $17.17$ | $13.05$ |
> > | L-LAS-AT (Ours) | $38.54$ | $\mathbf{19.95}$ | $\mathbf{19.73} $| $\mathbf{19.67} $ | $\mathbf{14.33}$ |
> >
> > We appreciate the time and effort that you have dedicated to reviewing our manuscript. Just a quick reminder that discussion stage 1 is closing soon.
> >
> > Have our responses addressed your major concerns?
> >
> > If there is anything unclear, we will address it further. We look forward to your feedback.
> >
> > [B] Jia X, Zhang Y, Wu B, et al. LAS-AT: adversarial training with learnable attack strategy. CVPR 2022.
> >
> > Best, Authors of Paper 1779

---

### Official Review · Reviewer_EcMM · 2023-11-10

**Soundness:** 3 good
**Presentation:** 3 good
**Contribution:** 3 good
**Rating:** 6
**Confidence:** 4

**Summary:**

The authors present a two-stage update rule for adversarial training, resulting in a general adversarial defense framework.

**Strengths:**

The motivation is clearly explained.
The paper is well-written.
Introducing historical state of the target model as its proxy to construct a two-stage adversarial defense framework, is a novel idea.

**Weaknesses:**

It would be interesting to see results on out-of-distribution samples.

**Questions:**

Please see above.

---

> ### Author Response · Authors · 2023-11-13
> **To Reviewer #1**
>
> We sincerely appreciate your acknowledgment of the contributions and innovative aspects of our paper, as well as your attention to the motivation behind our proposed method. Your recognition is of great significance to us and serves as a tremendous source of encouragement.  Answers to specific points are below.
>
> **Q1.** *``It would be interesting to see results on out-of-distribution samples.''*
>
> **R1.** Thanks for the suggestion. We acknowledge the importance of exploring the generalizability of our proposed framework. In the next revised manuscript, we will include experiments and results specifically focused on out-of-distribution samples to provide more comprehensive evaluations of our framework. Due to the limited time available for rebuttal, we will give priority to verifying the experiments you mentioned based on the CIFAR10 dataset and used the trained models in the experiments. Specifically, we plan to generate out-of-distribution-samples with the mixup techniques, and test the performance based on the pretrained model in the experimental part. We will try our best to report new experimental results across different domains in the reply as soon as possible.
>
> Concerning your question, we would appreciate further clarification to ensure that we address your concerns adequately. Please provide specific details or elaborate on the areas where you seek additional information, and we will ensure to address them thoroughly in our revision. Once again, thank you for your constructive feedback. We are committed to enhancing the quality of our manuscript based on your suggestions, and submit a revised version that addresses all concerns.

---

> > ### Author Response · Authors · 2023-11-15
> > **Results on out-of-distribution samples**
> >
> > Regarding the experiments on out-of-distribution samples, we utilized the PreActResNet-18 model trained on the current classification dataset. Employing mixup on the validation dataset of the CIFAR10 dataset allowed us to generate new out-of-distribution data samples. The experimental results for both Standard Adversarial Training (SAT) methods and our enhanced approaches are summarized in the table below. As evident from the table, our defense framework exhibits superior robustness against attacks induced by out-of-distribution samples. If you have any further inquiries or suggestions, we welcome continued discussion on this matter.
> > |PreActResNet-18 | CIFAR10 dataset | PGD-10   | CIFAR10 dataset | PGD-50  |
> > | :---: | :--------: | :---:  | :--------: | :---: |
> > |  Method  | $\boldsymbol{\epsilon}=8 / 255$ | $\boldsymbol{\epsilon}=16 / 255$  | $\boldsymbol{\epsilon}=8 / 255$ | $\boldsymbol{\epsilon}=16 / 255$ |
> > | Fast-AT | $46.14$ | $14.46$ | $43.95$ | $9.25$ |
> > | LFast-AT (Ours) | $\mathbf{46.37}$ | $\mathbf{14.77} $| $\mathbf{44.79} $ | $\mathbf{10.33}$ |
> > | Fast-AT-GA | $48.27$ | $16.52$ | $45.76$ | $11.48$ |
> > | LFast-AT-GA (Ours) | $\mathbf{48.65}$ | $\mathbf{17.26} $| $\mathbf{46.57} $ | $\mathbf{12.73}$ |
> > | Fast-BAT | $49.24$ | $18.41$ | $47.94$ | $13.64$ |
> > | LFast-BAT (Ours) | $\mathbf{49.58}$ | $\mathbf{19.69} $| $\mathbf{48.75} $ | $\mathbf{15.48}$ |
> > | PGD-AT | $49.53$ | $18.57$ | $47.88$ | $13.64$ |
> > | LPGD-AT (Ours) | $\mathbf{51.87}$ | $\mathbf{21.90} $| $\mathbf{50.96} $ | $\mathbf{17.70}$ |

---

### Author Response · Authors · 2023-11-22
**Message to all reviewers**

Dear reviewers,

Thank you for all of your insightful comments and constructive suggestions. We greatly appreciate the recognition from the majority of the reviewers that **our work is relevant to the community and that the introduced proxy model and the proposed framework are novel and effective in improving the robustness accuracy and stability of adversarial training**.

We highly value the detailed comments, insightful suggestions, and constructive criticism provided by the reviewers. Your feedback will undoubtedly play a crucial role in refining and improving our work. During the rebuttal period, according to the suggestions of four reviewers, we tried our best to provide detailed results, such as **results on out of distribution samples**, **performance under L2 norm attacks**, **results on the larger dataset and comparison with more multi-step methods**, to further improve the generalizability and consistent improvement of the defense framework. We assure you that we will carefully address the the issues raised during our discussions, such as unclear visualizations and descriptions of methods and any other concerns that were mentioned. As described in the contribution part, **this paper aims to construct a simple but much effective two-stage adversarial defense framework**. And **the core of the experiment design is to demonstrate the consistent performance improvement of LAST framework on various existing SAT methods, which can be observed by the smoother adversarial landscapes, more stable training curves, higher mean robust accuracies and smaller standard deviations.**

In our subsequent version, we will make necessary revisions to ensure that the clarity and quality of our visualizations and presentation are improved. We will also **consider introducing more techniques to address the tradeoff between clean accuracy and robust accuracy, and provide more results on SOTA methods and datasets**. Once again, we sincerely thank the reviewers for their valuable input, which has been instrumental in enhancing the overall quality of our work. Your feedback is greatly appreciated, and we are committed to incorporating these improvements into our final version.

Best,

Authors